# Data Compressibility Quantifies LLM Memorization

**Yizhan Huang**                                                    *yzhuang22@cse.cuhk.edu.hk*
*The Chinese University of Hong Kong*

**Zhe Yang**                                                              *zhe012@e.ntu.edu.sg*
*Nanyang Technological University*

**Meifang Chen**                                         *1155173961@link.cuhk.edu.hk*
*The Chinese University of Hong Kong*

**Huang Nianchen**                                                    *huangnia@usc.edu*
*University of Southern California*

**Jianping Zhang**[*]                                           *jianpingzhang@meta.com*
*Meta*

**Michael R. Lyu**                                                    *lyu@cse.cuhk.edu.hk*
*The Chinese University of Hong Kong*

**Reviewed on OpenReview:** *https://openreview.net/forum?id=6L4UXc7P3h*

## Abstract

Large Language Models (LLMs) are known to memorize portions of their training data and sometimes reproduce content verbatim when appropriately prompted. Despite substantial interest, existing LLM memorization research has offered limited insight into how training data influences memorization and largely lacks quantitative characterization. In this work, we build upon the line of research that seeks to quantify memorization through data compressibility. We analyze why prior attempts fail to yield a reliable quantitative measure and show that a surprisingly simple shift from instance-level to set-level metrics uncovers a robust empirical phenomenon, which we term the *Entropy–Memorization (EM) Linearity*. This relationship establishes that a set-level data entropy estimator exhibits a linear correlation with memorization scores.

We validate our findings through extensive experiments across a wide range of open-source models and experimental configurations. We further investigate the role of the token space—an implicit yet pivotal factor in our method—and identify a variant of the EM Linearity. In addition, we make a side observation that our finding enables a simple application to distinguish between LLM train data and test data.

## 1 Introduction

Large Language Models (LLMs) are shown to memorize and reproduce verbatim sequences from their training corpora (Carlini et al., 2019; 2020). Such memorization behavior has raised growing concerns, particularly regarding privacy leakage and intellectual property protection. For example, studies have shown that LLMs can inadvertently generate personally identifiable information (PII) (Carlini et al., 2020), or proprietary data from books (USAuthorsGuild, 2023; LLMLitigation, 2023; Cooper et al., 2025) and news articles (Michael, 2023). Most recently, Anthropic reached a USD 1.5 billion settlement with authors over the unauthorized use of copyrighted books, underscoring the growing legal risks surrounding LLM training data (The New York Times, 2025).

---

[*]Corresponding author. Work done in CUHK.

As the scaling law (Kaplan et al., 2020) drives LLM developers to expand model capacity and training data for performance improvements, research (Wang et al., 2025; Ippolito et al., 2023) has demonstrated that memorization scales with model size. Broader data exposure in LLM training elevates the risk of leakage for all internet-sourced content. Therefore, advancing the theoretical understanding of the factors that shape memorization has become a crucial and urgent issue in LLM development. In general, factors can be categorized into three types: prompting strategy (Carlini et al., 2020; Schwarzschild et al., 2024), model training (Chu et al., 2025), and training data.

The existing memorization literature is limited in two respects. *First*, comparing the first two factors, the role of training data in memorization is under-explored. Existing research limits the scope to data duplication, where researchers find out that data duplication significantly increases memorization (Kandpal et al., 2022; Biderman et al., 2023b). Beyond that, with the belief that highly compressible text is easier to memorize, some attempts have been made to link training data compressibility to memorization (Carlini et al., 2020; Prashanth et al., 2025). However, such attempts fail to identify significant patterns between compressibility and memorization. *Second*, most existing memorization explorations are limited to *qualitative* studies. Most research work typically regards memorization within a binary framework (Carlini et al., 2020; Zhang et al., 2023; Prashanth et al., 2025; Schwarzschild et al., 2024). The only quantitative studies are by Carlini et al. (2023) and Zhou et al. (2024), and Morris et al. (2025), which investigate model scale, data duplication, and context length.

Concerning the above limitations, this paper tackles an open question: **How to characterize memorization by the compressibility of training data in LLMs quantitatively?** Previous quantitative studies fail to address this. This paper formulates memorization using an integer-valued memorization score (between LLM response and golden answer). In this work, we adopt two metrics related to compressibility: 1) zlib compression, which is inspired by prior work (Deutsch & Gailly, 1996; Carlini et al., 2020) and 2) an entropy (estimator) motivated by their theoretical connection with compressibility in information theory.

This paper identifies a significant defect of previous attempts: metrics are evaluated *instance*-wise. Each instance provides limited and noisy information, since the underlying token space is much smaller than the overall token space. Motivated by this, we instead work on *set*-level. Our set-level approach is also inspired by recent "dataset inference", or set-level membership inference attack (MIA) approaches (Maini et al., 2021; 2024), where researchers found that instance-level MIA shows limited robustness and works on set-level approaches.

With a simple modification from instance-level approaches to set-level, we demonstrate that the set-level entropy estimator accurately approximates the memorization score. Measuring fitness using linear regression, we achieve the Pearson Correlation $r > 0.9$ across a wide range of popular LLMs. We dub this core finding of the study as **Entropy–Memorization Linearity**. It suggests that training sequences with higher entropy are strongly correlated with higher memorization scores (i.e., lower proximity between the model's response and ground truth data itself). Such linear correlation is empirically validated on a wide range of pre-trained models, including the OLMo family (Groeneveld et al., 2024), OpenLlama (Geng & Liu, 2023), and Pythia (Biderman et al., 2023b). We also explore EM Linearity in various experimental setups, including continuation length and inference sampling strategy.

Our entropy estimator enjoys twofold benefits: 1) the metric gives a *quantitative* description of memorization. The quantitative metric advances beyond the traditional binary setting of memorization with qualitative empirical observations. A quantitative metric facilitates the assessment of privacy risks for LLM providers. 2) the metric is *model-agnostic*. A model-agnostic approach is compute-efficient. It does not require backpropagation with a large number of model weights. In contrast, model-aware approaches, such as influence functions (Koh & Liang, 2017; Feldman & Zhang, 2020), typically require high computation like Hessian operation or retraining on LLMs.

We conduct thorough investigations into EM Linearity under several dimensions. **First**, we consider an implicit factor that shapes EM Linearity: token space, the support set over which entropy is defined. We identify that lower memorization-score data comprises *exponentially-linear* fewer unique tokens, and achieves *linearly* higher entropy values given the support size. **Second**, by applying our findings to addtional test

data, we uncover a simple yet effective method for distinguishing training data from test data. This leads to a simple dataset inference attack, enabling privacy auditing.

To summarize, our EM Linearity advances beyond existing research in the following ways:

- For the first time, we step beyond *instance*-level statistics in LLM memorization and obtain a *set*-level statistics that approximates LLM memorization well (§ 3). We term this as Entropy–Memorization Linearity. This pattern is preserved under various experimental setups (§ 4 and Appendix C and D).

- Different from previous measures that heavily depend on model or prompting, our set-based entropy estimator is characterized by the training data. To understand this data-centric metric, we explore the token space, an implicit factor within the training data, and identify a variant of EM Linearity (§ 5).

- Our set-based entropy estimator provides a *quantitative* characterization of memorization. Leveraging this quantitative nature—which most prior work lacks—we further show that the EM Linearity naturally induces a simple dataset inference attack, enabling practical auditing of privacy risks (§ 6).

## 2    Preliminaries

To provide background for the study, we first review the efforts to link compressibility to memorization in Section 2.1. Then in Section 2.2, we establish the required notation and explain the experimental setup that we follow throughout the paper.

### 2.1    LLM Memorization and its link to compressibility

Prior work on characterizing memorization can be broadly grouped into three categories: (i) model-centric factors, such as training paradigms and model scale (e.g., larger models tend to memorize more (Ippolito et al., 2023)); (ii) prompt-centric factors, such as prompting strategies (e.g., longer prompts tend to elicit more memorization (Carlini et al., 2023)); and (iii) data-centric factors, which focus on properties of the training data itself. For a comprehensive review, we refer readers to Section 7.

Regarding data-centric factors, a line of seminal work (Kandpal et al., 2022; Carlini et al., 2023) proposes that data repetition significantly increases the chances of memorization. Another line of research tries to link data compressibility to memorization based on an implicit suspicion in the community: highly compressible text corresponds to lower memorization difficulty. Such efforts started from the very first LLM data reconstruction work done by Carlini et al. (2020), where researchers use zlib compression to filter out repetitive text. Later, under a binary framework of memorization (i.e., memorization *v.s.* non-memorization), Prashanth et al. (2025) measured the link between text compressibility and memorization, but they did not observe a significant pattern. Note that perplexity, as explored by previous studies (Aerni et al., 2025; Prashanth et al., 2025; Huang et al., 2024), does not fall under the scope of data-centric factors, since it measures the uncertainty of *models*. So far, researchers have yet to establish a satisfactory connection between memorization and compressibility.

**Metrics of compressibility**   In this paper, we adopt two compressibility metrics: 1) zlib compression ratio (Deutsch & Gailly, 1996), and 2) estimated entropy [1] based on empirical point probabilities (Carlton, 1969). The use of zlib compression follows Carlini et al. (2020); while the second entropy-based metric is inspired by the close relationship of compressibility and entropy proved by the source coding theorem (Shannon, 2001). Note that two metrics correspond to two types of lossless coding algorithms in information theory: coding schemes for sources with memory and without memory, i.e., memoryless.

**Previous work fails to characterize memorization through data compressibility.**   Since prior work explores memorization with different experiment setups, we reproduce the result under a unified setup.

---

[1]We assume a base-2 logarithm for all entropy calculations throughout the work.

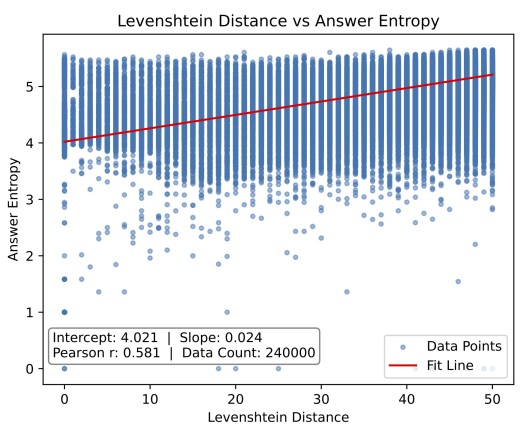
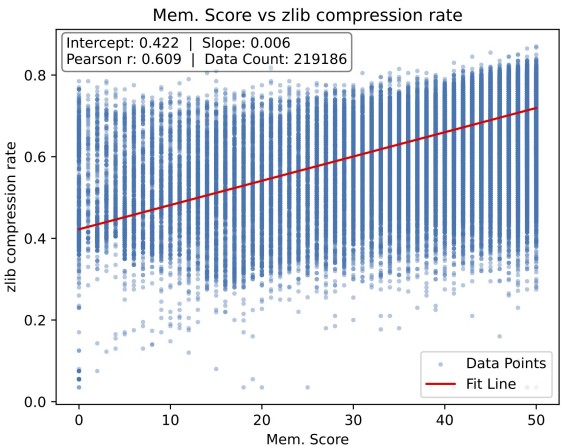

(a) entropy estimator *v.s.* memorization score.

(b) zlib compression rate *v.s.* memorization score.

Figure 1: Existing compressibility metrics fail to capture memorization score on OLMo-1B. In this figure, each point represents a sampled text sequence from the LLM training corpus, and sampling is repeated $N = 240,000$ times.

Figure 1 presents the result of instance-level entropy estimator and zlib compression ratio. The algorithms are detailed in the Appendix B, and the experimental setup will be explained in Section 2.2. In general, we observe a positive, yet weak, linear Pearson correlation, with $r$ around 0.6.

## 2.2 Experimental Setup

**Threat Model**  This paper assumes a hypothetical engineer who studies the characterization of training data on an LLM. Therefore, it is necessary for the engineer to have full access to the LLM *and* its training data. This engineer controls for other potential confounders in the memorization score, including prompt strategy and training paradigm.

Consistent with the prevailing literature on memorization, this work focuses on pre-trained ("base") LLMs optimized via cross-entropy loss. We also evaluate post-trained models (e.g., "instruct" variants) in the appendix C.6.

**Choices of LLM and Training Corpus**  We selected four LLMs: (1) OLMo (Groeneveld et al., 2024) pre-trained on Dolma (Soldaini et al., 2024) dataset, and (2) OLMo-2 (OLMo et al., 2024) pre-trained on OLMo-2-1124-Mix (OLMo et al., 2024) dataset; (3) OpenLlama (Geng & Liu, 2023) pre-trained on Redpajama (Computer, 2023); (4) Pythia (Biderman et al., 2023b) pre-trained on the Pile (Gao et al., 2020).

**Prompting Strategies**  In this study, we consider a *Discoverable Memorization* (DM) scenario (Nasr et al., 2025; Carlini et al., 2023; Kandpal et al., 2022; Ippolito et al., 2023). Formally, DM denotes the following: we sample $N$ token sequences from the training dataset. Each sequence is partitioned into $(p, s)$, where $p$ (first $|p|$ tokens) serves as the prompt and $s$ (remaining tokens) serves as the answer for model $\theta$. Afterwards, LLM $\theta$ generates a response $r = \theta(p)$. The memorization score measures the difference between two sequences $r$ and $s$. By default, we set $|p| = 100$ and $|s| = |r| = 50$, following the popular setup in the community (Al-Kaswan et al., 2023). However, variants of continuation length will be discussed in Section 4. We defer the detailed statistics of the sampled datasets to Appendix A.2. The token sequence sampling strategy is as follows: we repeatedly randomly sample a sequence (with length $= |p + s|$) from the dataset until the number reaches the required number.

**Filtering Trivial Memorization** We exclude *trivial* memorization cases where the model's response $r$ exhibits high lexical overlap with the prompt $p$. For example, the LLM may copy a long URL from the prompt to its response. Such cases are outside the scope of our interest, as memorization is shaped by the prompt rather than the training data. We devise a Longest Common Subsequence (LCS)-based filtering approach. We establish a thresholding strategy based on LCS: samples for which $LCS(p, s) \geq \frac{|s|}{2}$ are excluded from further memorization analysis, while samples below the threshold are retained.

**Memorization Score** We adopt the notion of memorization score $d(r, s)$ to measure the differences between response $r$ and answer $s$ at the token level. Following previous memorization work (Dong et al., 2024), we use Levenstein distance, or edit distance (Levenshtein et al., 1966), i.e., $d(r, s) = d_{\text{lev}}(r, s)$. It is defined by the minimal number of single-token edit operations – insertions, deletions, and substitutions – required to transform one sequence into another. A **higher** memorization score indicates **lower** similarity between two sequences. Such a definition offers two key properties to our memorization score: 1) It functions as an interval scale, allowing for a more nuanced quantification of the memorization; and 2) It captures verbatim sub-sequences even when they are offset by minor variations. A detailed discussion of the intuition underlying the memorization score is provided in Section A.1.

Note that we choose not to use a semantic-based memorization score, as semantic-level memorization does not have clear or direct social implications. For example, authors are suing LLM providers over the use of copyrighted books because model outputs exhibit substantial verbatim or near-verbatim overlap with their texts, not because the models generate passages that are merely semantically similar.

To summarize, the research question in this paper is formulated as follows:

> **Assumption.** A fixed pre-trained LLM $\theta$, a fixed prompting strategy $DM$ to generate $p$, LLM response $r = \theta(p)$, and a memorization score $d(r, s) = d_{\text{lev}}(r, s)$.
>
> **Goal of the study.** Find an approximator function $M(s)$ of memorization score $d(r, s)$.

## 3 Methodology

### 3.1 Limitation of prior instance-wise compressibility metrics

As demonstrated in Section 2.1, prior approaches fail to reliably approximate memorization. We attribute this limitation to the sparsity inherent in *instance-level* computation: a single sample covers only a negligible fraction of the full token space, rendering such metrics fundamentally incapable of capturing distributional characteristics. Formally, given a fixed memorization score $e$, the objective is to estimate a statistic (entropy) of a distribution conditioned on memorization score $e$, i.e., $p(s \mid e)$. Previous methods essentially treat a single instance $s_i = (s_i^1, s_i^2, \ldots, s_i^{|s_i|})$ as a sufficient representation of this conditional distribution $p(s \mid e)$, hence induces a large variance in estimation.

In practice, under our experimental setup, a single instance $s_i$ spans at most $|s_i| = 50$ tokens. This is orders of magnitude smaller than the full vocabulary $|\mathcal{T}|$. For example, for OLMo-1B, $|\mathcal{T}| \approx 50,000$, hence $|s_i|$ is the 0.1% of the available token space. Even saturating the context window limit (e.g., 4,000 tokens for OLMo-1B) does not bridge this gap. Consequently, instance-level compressibility measures remain highly noisy, failing to provide a robust estimate in real-world regimes where per-instance support is critically sparse.

### 3.2 Set-level compressibility metrics

In this work, we address the limitation of previous work by substantially expanding the size of the token space – We consider a "level-set" based method that uses all possible token samples with memorization score $e$. Specifically, we expand the token space from the tokens of *one* instance to *all* instances with the same memorization score $e$. In math notations, for a fixed memorization score $e$, the new token space is defined as:

$$\mathcal{T}_e = \bigcup \left\{ s_i^j \mid d(r_i, s_i) = e, i \in \{0, \cdots, N-1\}, j \in \{0, \cdots, |s|-1\} \right\}. \tag{1}$$

By adopting the above heuristics to construct set-level estimates, we aim to obtain a token space whose scale is comparable to that of the full vocabulary. Next, we describe how we implement the zlib-based and entropy-based methods at the set level.

**zlib method.** We adopt zlib compression for the concatenation of sequences with the same memorization score $e$:

$$s_e = \bigoplus \left\{ s_i^j \mid d(r_i, s_i) = e, i \in \{0, \cdots, N-1\}, j \in \{0, \cdots, |s|-1\} \right\}, \tag{2}$$

where $\oplus$ denotes string concatenation. Then, zlib-based compressibility is calculated by the compression rate of zlib:

$$M_{\text{zlib}}(s_e) = \frac{|\text{zlib}(s_e)|}{|s_e|}. \tag{3}$$

**Entropy method.** The empirical probabilities $\hat{p}_e(x)$ are now calculated within the new space $\mathcal{T}_e$:

$$\hat{p}_e(x) \triangleq \hat{p}(x|e) = \frac{1}{|s| \cdot |\{d(r_i, s_i) = e\}|} \left| \{(i, j) \mid s_i^j = x, d(r_i, s_i) = e\} \right|. \tag{4}$$

We then use new empirical probabilities to derive a new level-set-based entropy estimate to approximate the memorization score $e$.

$$M_{\text{ent}}(s_e) \triangleq - \sum_{x \in \mathcal{T}_e} \hat{p}_e(x) \log \hat{p}_e(x). \tag{5}$$

We provide the complete algorithm in Alg. 1.

---

**Algorithm 1:** Compute set-level compressibility metrics.

---
**Input:** LLM $\theta$, and its training corpus $D$.
**Output:** Plot of $(e, M(s_e))$
1 Sample $N$ prompt-answer pairs $\{(p_i, s_i)\}$ from $D$; // Section 2.2
2 **for** $i \leftarrow 0$ **to** $N - 1$ **do**
3     $r_i \leftarrow \theta(p_i)$ // Prompt LLM for response
4     $d(r_i, s_i) \leftarrow d_{\text{lev}}(r_i, s_i)$ // Memorization score
5 **end**
6 **for** $e \leftarrow 0$ **to** $|s| - 1$ **do**
    // zlib method
7     $s_e \leftarrow \bigoplus \left\{ s_i^j \mid d(r_i, s_i) = e, i \in \{0, \cdots, N-1\}, j \in \{0, \cdots, |s|-1\} \right\}$ // Prepare sequence. Ref: Eq. 2
8     $M_{\text{zlib}}(s_e) \leftarrow \frac{|\text{zlib}(s_e)|}{|s_e|}$ // Obtain zlib compression rate. Ref: Eq. 3
    // Entropy estimator method
9     $\hat{p}_e \leftarrow \frac{1}{N|s|} \left| \{(i, j) \mid s_i^j = x, d(r_i, s_i) = e\} \right|$ // Obtain empirical probabilities. Ref: Eq. 4
10     $M_{\text{ent}}(s_e) \leftarrow - \sum_{x \in \mathcal{T}_e} \hat{p}_e(x) \log \hat{p}_e(x)$ // Calculate Entropy. Ref: Eq. 5
11     Plot $(e, M_{\text{ent}}(s_e))$ and $(e, M_{\text{zlib}}(s_e))$.
12 **end**

---

**Note on other plausible set-level approaches.** In addition to our primary method, we further investigate a variation involving a more aggressive expansion of the token space. This approach, which we denote as $n$-width-binning memorization scoring, is detailed in Appendix C.4. Notably, our primary method represents the special case where $n = 1$ within this broader framework.

## 4 Entropy–Memorization Linearity

We run algorithm 1 on an extensive range of open-dataset LLMs and present the empirical results in Fig. 2. It turns out that the level-set-based entropy estimator is a better approximator of the memorization score than the zlib method.

We observe powerful linear empirical results ($r = 0.972$ and $0.945$ respectively) on both plots. It indicates that **the level-set-based entropy estimator is an effective *linear* approximation of memorization**

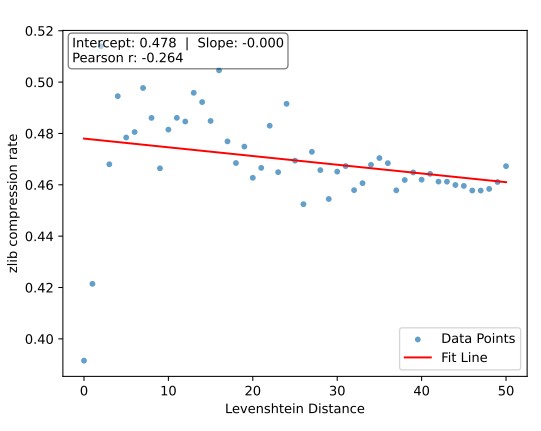

(a) zlib compression rate *v.s.* memorization score.

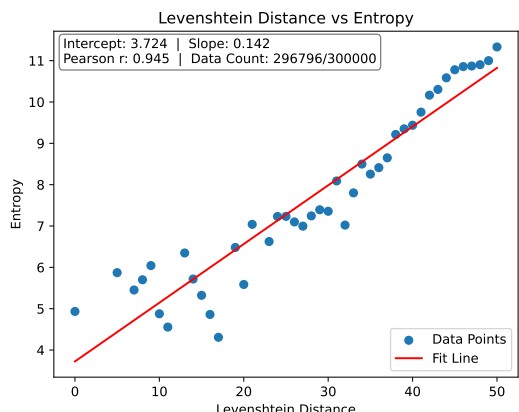

(b) Level-set-based entropy estimator *v.s.* memorization score.

Figure 2: At the set level, compared to zlib compression rate, entropy-based method achieves good approximation of memorization score. The experiments are conducted on OLMo-2-1124-7B.

**score**. We name this discovery as *Entropy-Memorization Linearity*. Here is a formal description of the linearity:

> **Entropy Memorization Linearity.** Given a fixed pre-trained LLM $\theta$, a fixed prompting strategy $DM$ to generate $p$, and a memorization score $d(r, s) = d_{\text{lev}}(\theta(p), s)$. $M_{\text{ent}}(s)$ serves as a linear approximator of the memorization score $d_{\text{lev}}(\theta(p), s)$.

In Figure 3, we provide extensive results on OpenLlama, Pythia-70m-deduped.

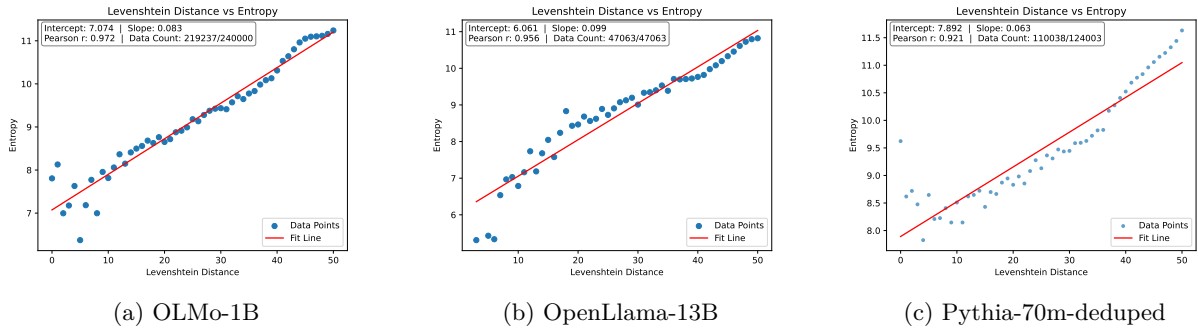

(a) OLMo-1B            (b) OpenLlama-13B            (c) Pythia-70m-deduped

Figure 3: Entropy–Memorization Linearity on open-dataset LLMs.

We explore various LLM inference sampling strategies, including temperature, top-p and top-k sampling in Appendix C.1. Experiments on a broader range of models are available in Appendix C.2. In appendix D, we also separate the whole datasets into $N$ semantic clusters, and then validate our finding within each cluster. Next, to showcase the validity of EM Linearity, we present our evaluation under varying continuation lengths.

**Entropy–Memorization Linearity is preserved under varying continuation lengths** We explore different continuation token lengths, including $\{10, 20, 30, 40, 50\}$. As a demonstration, we use OLMo-2-1124-7B and its training dataset OLMo-2-1124-Mix in this experiment. Note that we rescaled the memorization score to the range $[0, 1]$ in the plot for better presentation.

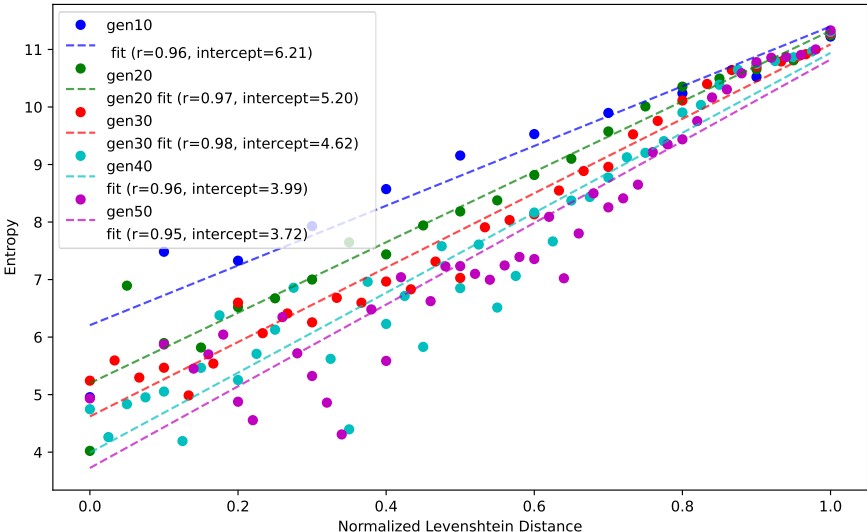

Figure 4: Entropy–Memorization Linearity under varying generation token lengths.

Figure 4 presents the experimental result when the generation token length varies. The Pearson correlation coefficients ($r$) remain very high (ranging from 0.92 to 0.98) across all settings, indicating robust fitness under varying token lengths. Although regression lines have different slopes and intercepts, the $y$-value reaches about 11 when the memorization score $e = 50$, i.e., $M(s_{50}) \approx 11$.

Another observation is that there is a monotonic increase in the intercept values of the fitted regression lines as the generation token length increases. Information theory may help to justify this: Denote the vocabulary size as $|\mathcal{T}|$, an $n$-sequence has $|\mathcal{T}|^n$ potential outcomes; hence, when $n$ gets larger, the maximum entropy of the $n$-sequence increases.

Besides, we observe that EM Linearity weakens under out-of-distribution data. We also explored a boundary case of EM Linearity, where the correlation weakens on a biomedical research data subset. We provide a detailed description in the Appendix C.5.

## 5 Normalized Entropy–Memorization Linearity

Our method effectively enlarges the token space and provides a reliable proxy for the memorization score. However, we have not yet conducted a detailed investigation of the underlying token space itself. In fact, token space size plays an implicit yet crucial role in EM Linearity. In information theory, the maximum entropy is bounded by outcome space cardinality. In fact, the maximum entropy of a discrete random variable is $-\sum_{i=1}^{n} \frac{1}{n} \log \frac{1}{n} = \log n$, where $n$ is the cardinality of the outcome space (i.e., token space size). The upper bound is achieved when the random variable follows a uniform distribution.

In this subsection, for each memorization score $e$, we report the corresponding token space size $\mathcal{T}_e$. The results are shown in Fig. 5a. The statistic is calculated following the same experimental setup on OLMo-2-1124-7B. The experimental result indicates that the token space size, or unique token count, grows exponentially as the memorization score increases. Low-score memorization occurs within a limited token space. Remarkably, while the full dataset contains tens of millions of tokens, perfect memorization (score=0) occurs within merely $2^{10}$ unique tokens, namely 1/50 of the vocabulary size. While we do not provide a further theoretical justification, we suspect that tokens appearing with higher frequency in the training corpus are more likely to be memorized. Further investigation may require another line of research: measuring memorization at the *token level*. In the context of membership-inference attacks, there are explorations like Tao & Shokri (2025).

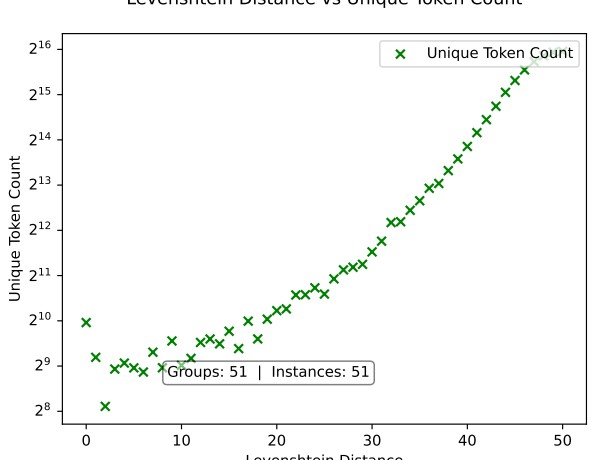
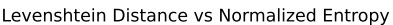
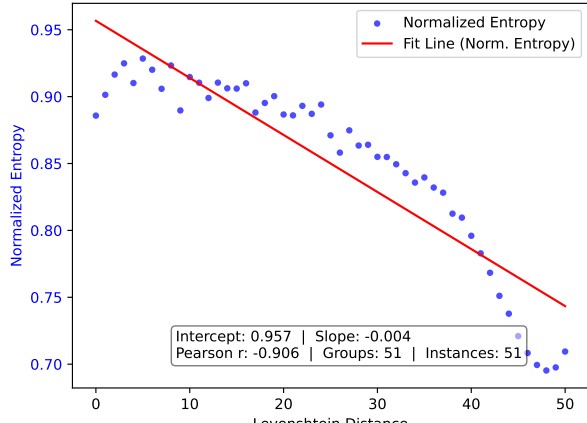

(a) Memorization score *v.s.* token space on OLMo-2-1124-7B. Note: the scale of the y-axis is *exponential*.

(b) Memorization score *v.s.* normalized entropy on OLMo-2-1124-7B.

Figure 5: Memorization score *v.s.* normalized entropy on OLMo-2-1124-7B.

Given a fixed outcome-space cardinality, our entropy estimator characterizes how the empirical probability mass is distributed. To factor out the influence of token-space size, we normalize the observed entropy by the maximum attainable entropy under the same token-space cardinality, and refer to the resulting ratio as the *normalized entropy*:

$$\overline{M}(s_e) \triangleq \frac{M(s_e)}{H_{\max,e}} = \frac{M(s_e)}{\log |\mathcal{T}_e|}, \tag{6}$$

where $\overline{M}(s_e)$ normalizes the entropy estimate $M(s_e)$ by its theoretical maximum $H_{\max,e}$. Values approaching 1 indicate a near-uniform distribution over the token set $\mathcal{T}_e$, while lower values suggest greater non-uniformity.

Following the same setup on OLMo-2-1124-7B, we plot $(e, \overline{M}(s_e))$ using blue dots on Figure 5b to observe how the normalized entropy estimator changes with the memorization score. Interestingly, we observe another linear trend – it indicates that normalized entropy (negatively) linearly decreases as the memorization score increases. In appendix C.3, we discuss the pattern with different sequence lengths.

Although a theoretical justification is not available for the phenomenon, a potential explanation is that high-memorization-score samples tend to follow a more "natural" language distribution, which is known to be long-tailed (Zipf, 2016) and therefore exhibits lower entropy. In contrast, low–memorization-score samples may contain more specialized content—such as code snippets or numerical values—whose distributions deviate from typical natural language regularities.

**Summary.** By decoupling the entropy estimator to outcome-space cardinality and randomness of distribution, we reveal that 1) lower memorization-score data comprises *exponentially-linear* fewer unique tokens, and 2) achieves *linearly* higher entropy values given the outcome space.

# 6 Application: Dataset Inference using EM Linearity

## 6.1 Entropy Memorization Linearity on Test Data

Running algorithm 1 on *training* dataset of LLMs, we discovered the Entropy-Memorization Linearity. This section then explores another question that naturally arises – what happens if we run the same algorithm on *test* dataset? It turns out that the plot behaves very differently from training datasets.

Figure 6 presents the plot of applying the entropy estimator and the normalized entropy estimator to test data. To maintain consistency throughout the paper, we use the terms used for memorization with slight abuse. For example, the memorization score still measures the distance between the ground truth and the model's response, but the model is not actually "memorizing" training data. Following the same setup described in the main body, we select data from LiveBench (White et al., 2025) from 2024-06-25 to 2024-11-25. The time property guarantees that LiveBench is non-member data for OLMo-2-1124-7B. LiveBench is licensed under the CC BY-SA 4.0 International License.

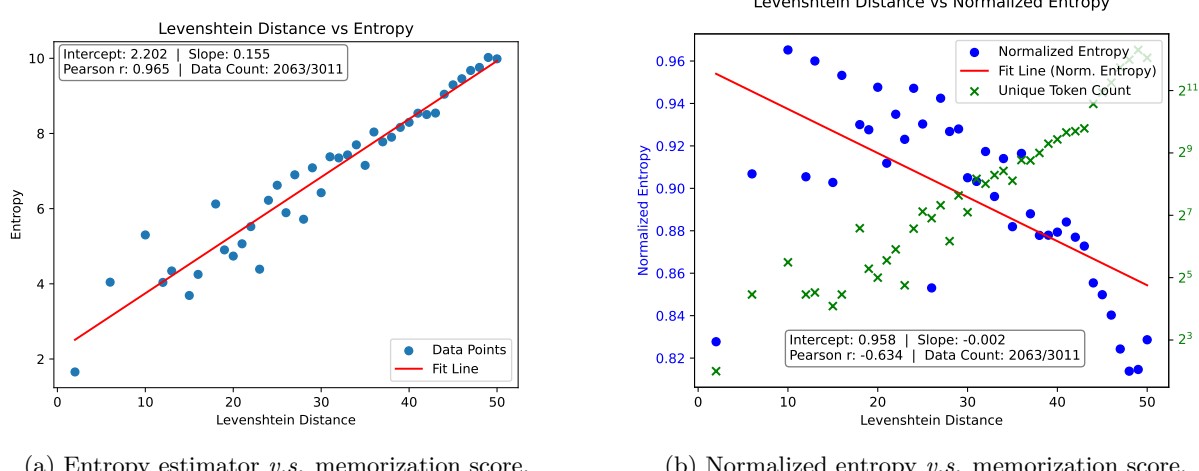

(a) Entropy estimator *v.s.* memorization score.

(b) Normalized entropy *v.s.* memorization score.

Figure 6: Entropy–Memorization Linearity on OLMo-2 with LiveBench dataset.

With around 3k samples from LiveBench, we observe that: (1) Level-set-based entropy is still a good indicator of LLM performance, as demonstrated by the high Pearson's r. The sample space also approximately grows exponentially as the memorization score increases. (2) Compared to the training data, the intercept is lower, and the slope is higher. There is less coverage of low memorization scores, especially for memorization scores within the range of 0-10. (3) In the low-distance set, it is observed that OLMo-2 produces low-entropy text. The entropy is much lower than what we observed in the train data.

Here is a case study on a 2-distance memorization:

**Prompt.** Please create a valid join mapping between CSV Table A and CSV Table B. Each column in A maps to 0 or 1 columns in B. Return your response as a Python dictionary, formatted as col_name_in_df_a : col_name_in_df_b. Please return only the dictionary.
CSV Table A: Areas,freq_1,freq_2,freq_3,freq_4,freq_5,freq_6 0.0,0.0,0
**Response.**
.0,0.0,0.0,0.0,0.0
1.0,0.0,0.0,0.0,0.0,0.0,0.0
2.0

Given the prompt in this case study, OLMo-2 generates repeated numbers, exhibiting low entropy. Based on the observations above, we derive a simple strategy to distinguish between train data and test data for LLM.

## 6.2 Data Inference Methodology

**Dataset Inferences** Dataset Inference (DI) (Maini et al., 2024; 2021) builds on the idea of membership inference attacks (MIA) (Song & Shmatikov, 2019; Galli et al., 2024; Carlini et al., 2022; Mattern et al., 2023; Jagannatha et al., 2021). Both MIA and DI aim to identify whether some suspect data was part of the training data, but they differ on the amount of data required. MIA operates at *instance* (sentence) level; however, DI operates on a *collection* of instances – in reality, the suspect data used for DI could be a book.

In practice, DI can identify potential test set contamination to provide a calibrated performance evaluation of LLMs. DI may also detect unauthorized usage of copyrighted training data, thus promoting the protection of intellectual properties. Moreover, by using more data to determine membership, DI is deemed to be more realistic than MIAs. As several works Duan et al. (2024); Maini et al. (2024); Hayes et al. presented, as the size of the training set increases, the success of membership inference degrades to random chance. Our DI method is inspired by several empirical observations. First, over-parametrization of LLMs may lead to overfitting on training data, resulting in a generalization gap between training data and testing data; then, LLMs may perform well on low-entropy testing data, resulting in a low intercept in EM Linearity. Besides, given a fixed dataset and LLM, empirical evidence suggests that the intercept and slope generated by Algorithm 1 are dependent. This inspires us to develop the following strategy for dataset inference:

> Given an LLM $\theta$ and dataset $D$, run Algorithm 1 and get intercept $k$. Compare $k$ with a pre-defined threshold $\tau_k$. Assign label 1 (i.e., member) if $k > \tau_k$, assign label 0 otherwise.

**Amount of data required for our DI method** Effective dataset inference requires reliable entropy estimation and diverse memorization score distributions. In the main body, we have revealed that the frequency of a low memorization score is exponentially smaller than that of a high memorization score. Therefore, we set the minimum sample size to $n = 1,500$, with each sample being a 150-token sequence. Appendix E.1 discusses how we determine the minimum size through a sensitivity test.

## 6.3 Experimental Results on Dataset Inference

Table 1: Dataset inference result

| LLM | Dataset | Intercept | Slope | Prediction | Ground-truth |
|---|---|---|---|---|---|
| OLMo-2 | LiveBench | 2.202 | 0.155 | 0 | 0 |
| Pythia | MIMIR_cc | -2.048 | 0.251 | 0 | 0 |
| Pythia | MIMIR_cc | 3.992 | 0.091 | 1 | 1 |
| OLMo-2 | OLMo-2-1124-Mix | 3.724 | 0.142 | 1 | 1 |
| Pythia | MIMIR_full | 6.297 | 0.092 | 1 | 0 |
| Pythia | MIMIR_full | 6.166 | 0.095 | 1 | 1 |
| Pythia | MIMIR_tarxiv | 1.156 | 0.174 | 1 | 1 |
| Pythia | MIMIR_tarxiv | -0.910 | 0.0227 | 0 | 0 |
| Pythia | MIMIR_wiki | 3.006 | 0.131 | 1 | 0 |
| Pythia | MIMIR_wiki | 2.894 | 0.133 | 1 | 1 |

*Selected LLMs.* We select OLMo-2-1124-7B ("OLMo-2"), Pythia-6.9B-deduped (Biderman et al., 2023b) ("Pythia").

*Selected Datasets.* We select LiveBench (White et al., 2025) and MIMIR (Duan et al., 2024). For LiveBench, we use data from 2024-06-25 to 2024-11-25. MIMIR is a public dataset originally for evaluating MIAs on the Pythia suite by re-compiling the Pile (Gao et al., 2020) train/test splits. For MIMIR, we use Pile CC ("cc"), temporal arXiv ("tarxiv"), "wiki" subset, and full dataset ("full") for evaluation. Note that LiveBench and Temporal arXiv are temporal-cutoff-based, while the remaining dataset is i.i.d.-based.

*Threshold for each LLM.* In our method, we assign a threshold for each LLM. We empirically set $\tau_k$ to 0 and 3 for Pythia and OLMo-2, respectively.

Table 1 presents the overall results of our method on the dataset inference task. In general, it achieves desirable accuracy. For more discussions on detailed comparison with baseline methods, we refer the reader to the appendix E.2.

### 6.4 Practical Use of the Dataset Inference Method

In practice, our DI method provides a practical tool for auditing the privacy of training data. An external auditor may apply the dataset inference method as follows:

1. Using a known subset of the training data, the auditor applies the set-level entropy method to obtain reference results. This setup is realistic, as only part of the training data is required.

2. The auditor determines an appropriate intercept threshold based on step 1.

3. The auditor collects a candidate dataset that is pending for dataset inference audit, and runs the set-level entropy method on the dataset.

4. Using the threshold from Step 2, the auditor makes a binary decision on the dataset membership.

**Advantages of our data inference method** It is compute-efficient – it only requires LLM inference on $n$ samples. It does not require any additional shadow or reference models.

## 7 Related Work

Since the discovery of the memorization phenomenon in the late 2010s (Zhang et al., 2017; Carlini et al., 2019; 2020; Feldman, 2020), the AI Security and Privacy research community has maintained a strong interest in the phenomenon and its implications. The following paragraphs examine how memorization in language models is influenced by key factors, including *training data, model paradigm, and prompting strategy.*

*Data shapes memorization.* Several studies suggest that (Kandpal et al., 2022; Biderman et al., 2023b) duplicated data significantly increases memorization. Larger models trained on larger datasets show increased memorization (Biderman et al., 2023a;b). Other studies (Tirumala et al., 2022; Wang et al., 2025) investigate how memorization manifests across data with varying semantics and sources.

*Model Paradigm shapes memorization.* Beyond pre-trained language models, recent work has explored memorization in post-training stages. Chu et al. (2025) demonstrate that supervised fine-tuned (SFT) LLMs exhibit stronger memorization tendencies than those trained with reinforcement learning (RL). Additionally, Nasr et al. (2025) reveals that safety-aligned models still retain memorized data.

*Prompting Strategy shapes memorization.* Researchers employ three main types of prompting strategies for language models, categorized by threat models. A significant body of work relies on manual efforts or template-based approaches to generate prompts at scale, as seen in Carlini et al. (2019; 2020); Kim et al. (2023). Studies such as Carlini et al. (2020); Kandpal et al. (2022); McCoy et al. (2021) demonstrate that longer prompts substantially increase the likelihood of reproducing memorized training data sequences. Another line of research constructs prompts directly from existing data sources, such as training corpora or web data (Nasr et al., 2025; Carlini et al., 2023; Kandpal et al., 2022; Ippolito et al., 2023; Aerni et al., 2025). Recent advances involve more sophisticated strategies that leverage synergies between LLMs and training data. For example, Zhang et al. (2023) quantifies how the performance of a model depends on whether an example $x$ was included in the training data. Additionally, Schwarzschild et al. (2024) adapts a prompt optimization tool to generate effective extraction prompts.

**Comparison with other qualitative studies.** While established literature Carlini et al. (2023) Zhou et al. (2024), and Morris et al. (2025) examine memorization quantitatively, our work differs from them, focusing on training data compressibility rather than factors such as model scale, data duplication, and context length.

## 8 Limitations and Broader Impacts

**Limitations**  *Predictive power of the Entropy-Memorization Linearity* Due to the limitation of the sample space as discussed in Section 3, our strategy does not enable memorization score prediction at the instance level. However, we see promising results on set-level memorization-related tasks, such as Dataset Inference.

*Empirical experiments.* Our work adopts a single set of prompting strategy (DM) and memorization score (edit distance) in our memorization experiments. Although the setup is commonly used by other studies, other combinations exist. We want to explore adversarial compression (Schwarzschild et al., 2024), and non-adversarial reproduction (Aerni et al., 2025) in our future work.

**Implications and Societal Impact**  For the research community, we believe the EM Linearity provides a useful foundation for theoretical understanding the factors that drive memorization in LLMs. Moreover, it enables a simple and scalable approach for privacy auditing. For practitioners training LLMs, our method allows pre-screening of training datasets to assess potential memorization risks. External auditors may also employ our dataset inference method to detect test-set contamination and identify potential copyright infringement issues in deployed models.

This paper uses existing open-research LLMs and their corresponding training datasets. To the best of the authors' knowledge, this research does not introduce any additional negative societal impacts.

## 9 Conclusions

This paper presents the Entropy-Memorization Linearity: a level-set-based entropy estimator of training data chunks linearly approximates the edit-distance-based memorization score. Further investigation indicates that this correlation is robust acrosss different sequence lengths, sampling strategies, and data clusters with different semantics. By examining vocabulary size, it is revealed that lower memorization-score data comprises *exponentially-linear* fewer unique tokens, and achieves *linearly* higher entropy values given the support size.

For future work, we plan to explore why the proposed level-set-based entropy estimator fits the memorization score so well. Potential theoretical tools include the long-tail theory of Feldman et al (Feldman, 2020; Feldman & Zhang, 2020), and multi-calibration in LLMs (Detommaso et al., 2024). Such efforts may also shed light on the interpretation of slope and intercept resulting from the EM Linearity.

## Acknowledgement

The work described in this paper was supported by the Research Grants Council of the Hong Kong Special Administrative Region, China (No. CUHK 14209124) of the General Research Fund, and RGC Grant for Theme-based Research Scheme Project (RGC Ref. No. T43-513/23-N).

We thank Farzan Farnia for his insightful comments and discussions.

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

# A  Details on Experimental Setup

## A.1  Justification on Memorization Score

We adopt a Levenshtein-distance-based memorization score. The score offers two key properties: (1) it functions as an interval scale, allowing for a more nuanced quantification of memorization; and (2) it captures verbatim sub-sequences even when they are offset by minor variations.

For instance, suppose the LLM continuation is "`The email is:  abc@example.com`" and the ground truth is "`The email is abc@example.com`". These sequences differ only by a single colon ("`:`"). Edit distance can still capture the match of the email address, since one operation (removing "`:`") leads to an exact match.

In contrast, position-dependent distance metrics, which compare strings index by index, fail to capture the shared email address substring. This makes our approach significantly more robust for detecting training data leakage in real-world, noisy scenarios.

### A.2 Composition of Sampled Dataset

We constructed datasets with sizes 240,000 and 300,000, respectively, for Dolma and OLMo-2-1124-Mix. Table 2 and 3 present the composition of sampled datasets after LCS filtering.

Table 3: Source Counts of OLMo-2-1124-Mix

| Dataset | Count |
|---|---|
| algebraic-stack | 797 |
| arxiv | 1,464 |
| dclm1 | 28,130 |
| dclm2 | 27,984 |
| dclm3 | 28,014 |
| dclm4 | 28,163 |
| dclm5 | 28,049 |
| dclm6 | 28,065 |
| dclm7 | 28,060 |
| dclm8 | 28,091 |
| dclm9 | 28,040 |
| dclm10 | 28,032 |
| open-web-math | 499 |
| pes2o | 4,364 |
| starcoder | 5,280 |
| wiki | 276 |
| Total | 293,308 |

Table 2: Source Counts of Dolma

| Source Dataset | Count |
|---|---|
| pes2o | 39,777 |
| cc | 39,743 |
| books | 39,598 |
| reddit | 39,373 |
| stack | 38,865 |
| wiki | 23,115 |
| Total | 219,186 |

## B  Implementation details of instance-wise compressibility metrics.

**Zlib method.**  For each instance $s$, we apply the zlib library to compress the sequence and report its compression rate, computed as the compressed length divided by the original length.

**Entropy method.**  An instance $s_i = (s_i^1, s_i^2, ..., s_i^{|s_i|})$ is a sequence, where $s_i^j$ is a token. All tokens within $s_i$ form the sample space $\mathcal{T}_i$. Then for each token $x \in \mathcal{T}_i$, the empirical point probabilities $\hat{p}_i(x)$ are calculated as:

$$\hat{p}_i(x) = \frac{1}{|s|} \left| \{j \mid s_i^j = x\} \right|. \tag{7}$$

$\hat{p}_i(x)$ is the relative frequency of $x$ in the observed sequence. In this attempt, we use entropy estimated by the empirical point probabilities as our approximator $M(s_i)$:

$$M(s_i) \triangleq - \sum_{x \in \mathcal{T}_i} \hat{p}_i(x) \log \hat{p}_i(x) \tag{8}$$

In practice, the distribution for language is unknown. We instead learn from samples. The above estimator approximates entropy by viewing the point probability as samples from the empirical distribution itself.

With the established $M(s_i)$, we are interested in whether $M(s_i)$ is a good approximator of the memorization score $d(r_i, s_i)$. To achieve this, we gather all $(M(s_i), d(r_i, s_i))$ pairs obtained by empirical observations in a scatter plot, and further study their correlation. The detailed algorithm is as follows:

---

**Algorithm 2:** Instance-wise entropy estimator.

**Input:** LLM $\theta$, and its training corpus $D$
**Output:** Plot of $(d(r_i, s_i), M(s_i))$
1  Sample $N$ prompt-answer pairs$\{(p_i, s_i)\}$ from $D$;
2  **for** $i \leftarrow 0$ **to** $N - 1$ **do**
3  $\quad$ $r_i \leftarrow \theta(p_i)$;
4  $\quad$ $\hat{p}_i \leftarrow \text{EmpProb}(r, s)$// `Eq. 7`
5  $\quad$ $M(s_i) \leftarrow - \sum_{x \in \mathcal{T}_i} \hat{p}_i(x) \log \hat{p}_i(x)$// `Eq. 8`
6  $\quad$ $d(r_i, s_i) \leftarrow d_{\text{lev}}(r_i, s_i)$
7  $\quad$ Plot $(d(r_i, s_i), M(s_i))$;
8  **end**

---

We run algorithm 2 on the OLMo-1B model, and obtain the scatter plot as illustrated in Fig. 1.

## C  Extended Results on Entropy-Memorization Linearity

### C.1  Additional Results with various sampling strategy of LLMs

In the main body of the paper, we assume a fixed temperature of 0.8. In this subsection, we adopt different sampling strategies of LLMs and discuss how these strategies might shape EM Linearity. Due to computation constraints, we conduct our experiments on a subset "DCLM1" with OLMo-2-1124-7B. The size of the subset is around 28,000.

We consider combinations of temperature, top-k sampling, and nucleus sampling (top-p). The experimental results are summarized in Tab. 4, and details are shown in Fig. 7 - 12. Under all sampling strategies that we have explored, we empirically observe that EM Linearity holds with $r > 0.92$. Beyond that, we make a few observations here:

- The zero-distance point $(0, M(s_0))$ exhibits a significant deviation from the regression line in both plots. When the memorization score $e = 50$, $M(s_{50}) \approx 11$.

- Intercept and slope are dependent if we fix the LLM and the dataset. The general pattern is that when the intercept increases, the slope decreases. This might indicate that the intercept and slope may have a degree of freedom 1.

- With a fixed temperature, enabling top-k or top-p sampling increases intercept and decreases slope.

- The estimated normalized entropy decreases with the memorization score increasing.

The first two observations are consistent with observation points 2 and 3 in Section 5.

Table 4: Entropy-Memorization Linearity under different LLM sampling strategy

| Strategy | r | Intercept | Slope |
|---|---|---|---|
| Temp=0 | 0.933 | 5.490 | 0.106 |
| Temp=0.5 | 0.936 | 5.474 | 0.106 |
| Temp=0.8 | 0.926 | 5.011 | 0.113 |
| Temp=0.8, top_p=0.5 | 0.935 | 5.599 | 0.103 |
| Temp=1 | 0.944 | 4.646 | 0.118 |
| temp=0.8, top_k=10 | 0.944 | 5.138 | 0.111 |

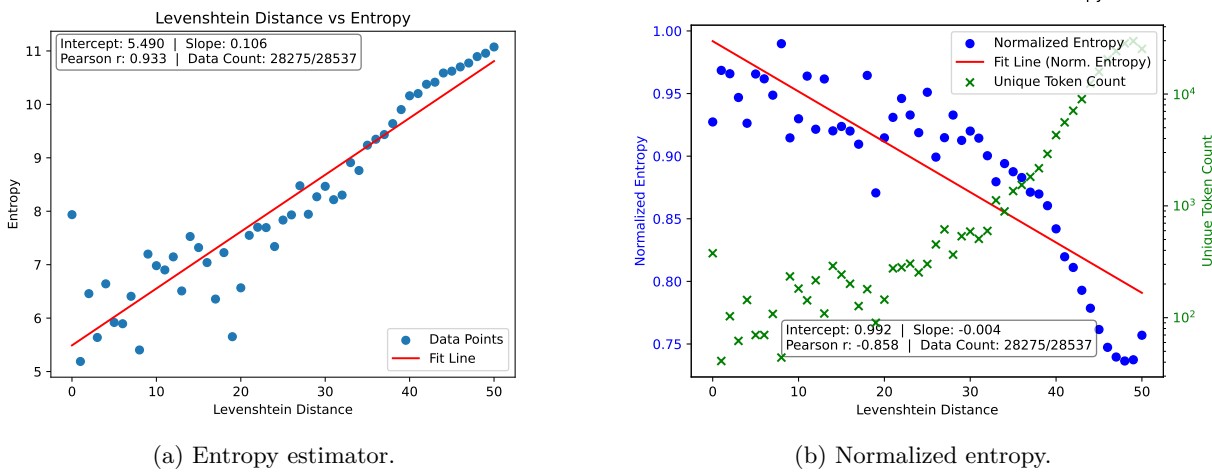

(a) Entropy estimator.

(b) Normalized entropy.

Figure 7: Sampling strategy: Temp=0.

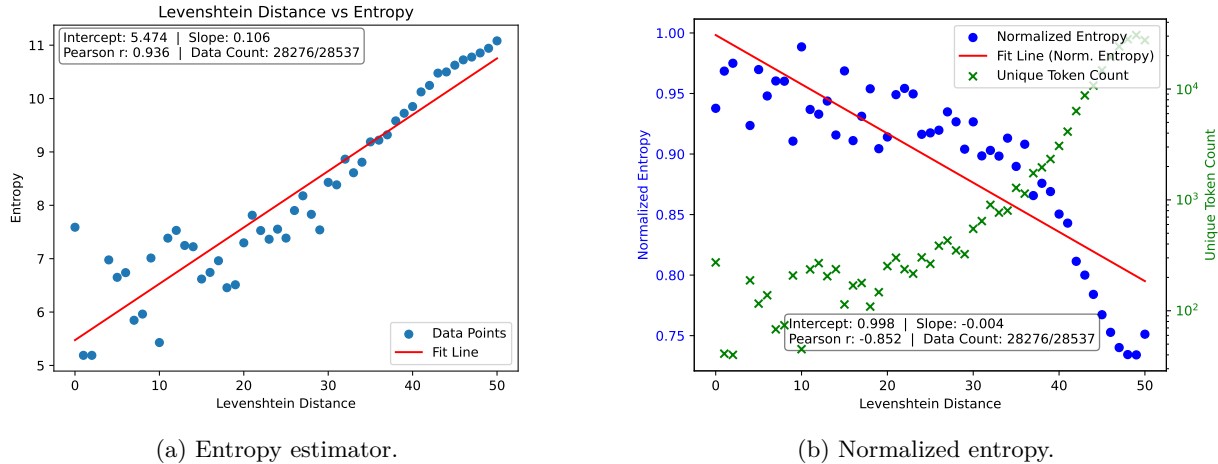

(a) Entropy estimator.

(b) Normalized entropy.

Figure 8: Sampling strategy: Temp=0.5.

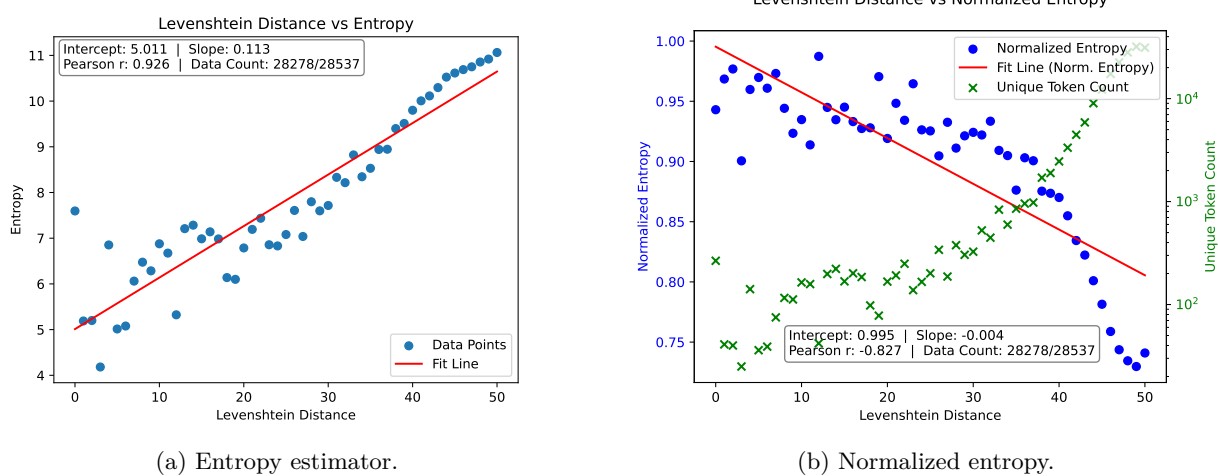

(a) Entropy estimator.

(b) Normalized entropy.

Figure 9: Sampling strategy: Temp=0.8.

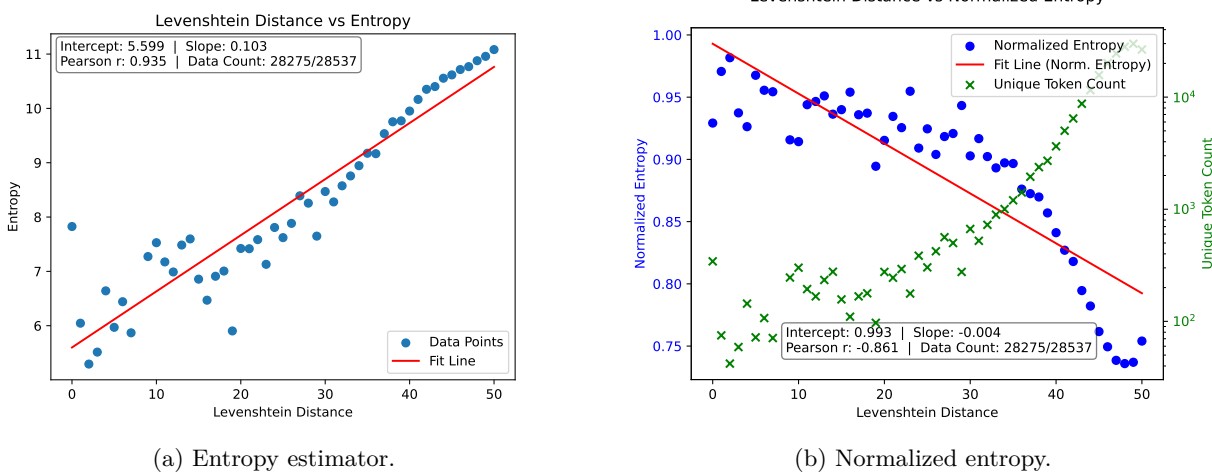

(a) Entropy estimator.

(b) Normalized entropy.

Figure 10: Sampling strategy: Temp=0.8, top-p=0.5.

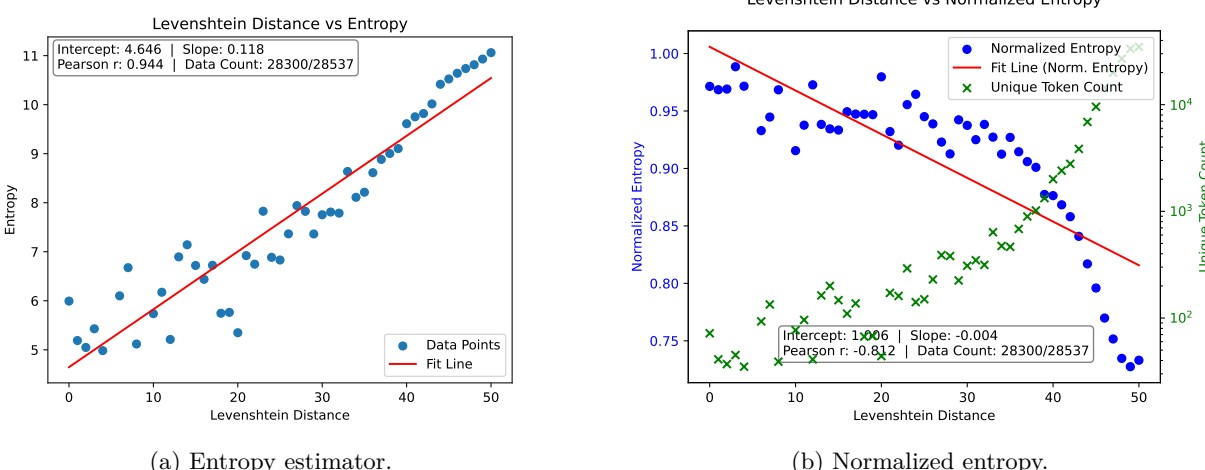

(a) Entropy estimator.

(b) Normalized entropy.

Figure 11: Sampling strategy: Temp=1.

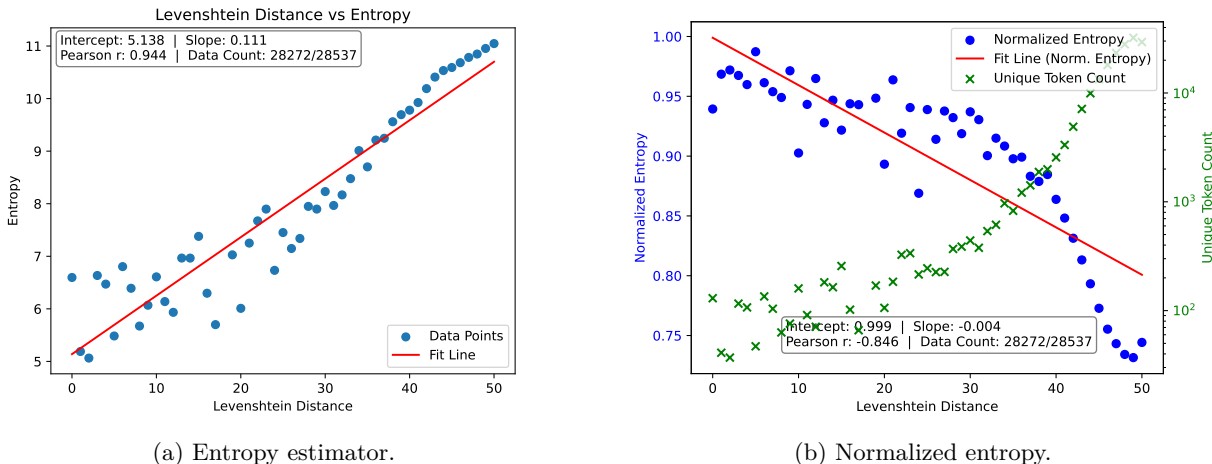

(a) Entropy estimator.

(b) Normalized entropy.

Figure 12: Sampling strategy: Temp=0.8, top-k=10.

## C.2 Entropy-Memorization Linearity

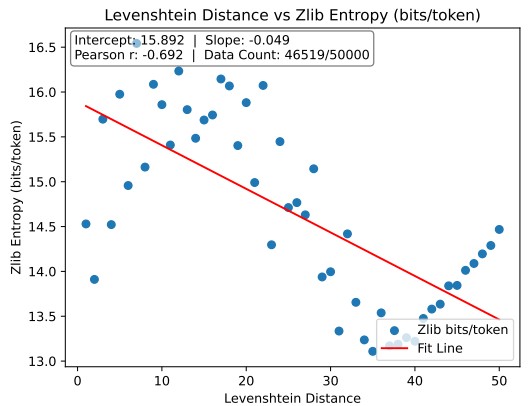
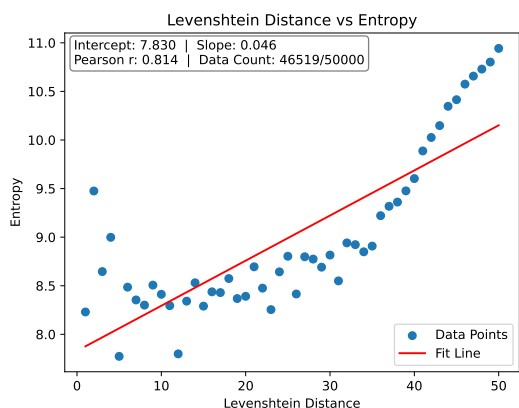

(a) Zlib compression rate *v.s.* memorization score for OpenLlama-7B.

(b) Level-set-based entropy estimate *v.s.* memorization score for OpenLlama-7B.

Figure 13: Comparison of compressibility- and entropy-based measurements of memorization for OpenLlama-7B.

The experimental results on OpenLlama-7B are shown in Figure 13a and 13b. The findings are consistent with the pattern observed in the main paper body.

## C.3 Normalized Entropy–Memorization Linearity

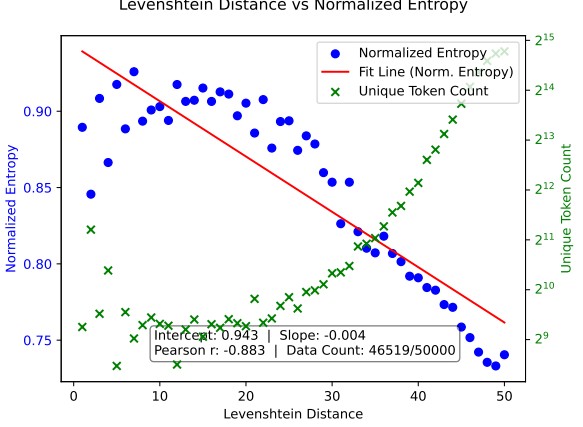
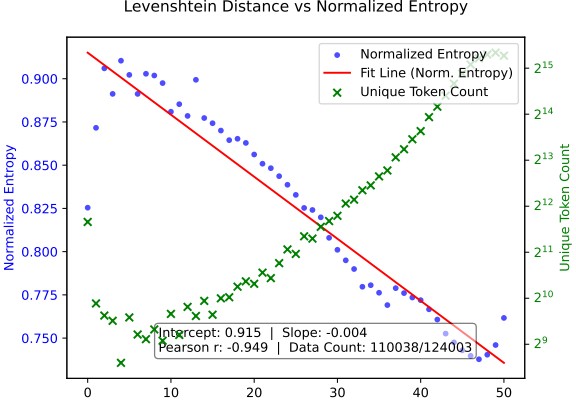

(a) Normalized Entropy–Memorization Linearity for OpenLlama-7B.

(b) Normalized Entropy–Memorization Linearity for Pythia-6.9B-dedup.

Figure 14: Normalized Entropy–Memorization Linearity across different model families.

Fig. 15 presents the (estimated) normalized entropy *v.s.* memorization score. The findings are consistent with the findings that we observed in the main paper body. Moreover, $|r|$ increases from 0.82 to 0.97 as generation length decreases.

## C.4 Entropy–Memorization Linearity under coarse binning of memorization score

In the main body of the paper, we adopt an **exact-score** binning method on memorization scores, i.e., we pool all tokens from the sequences within exactly the same memorization score. However, as we will show

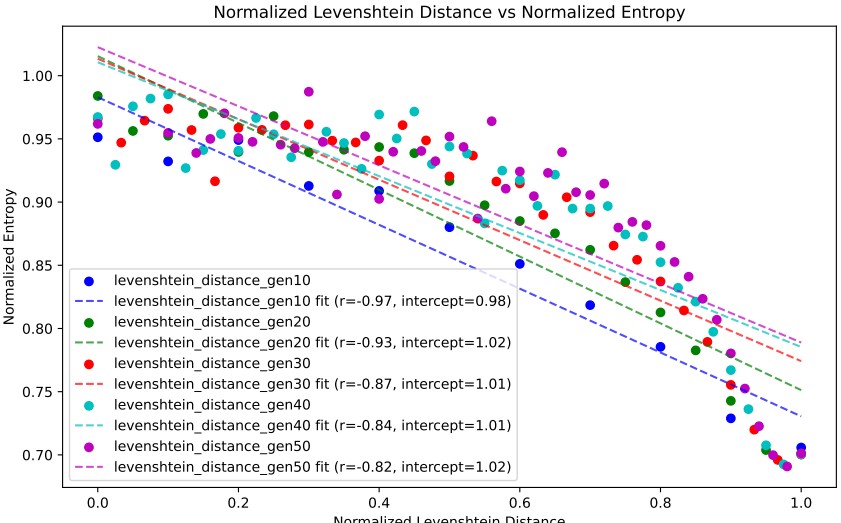

Figure 15: Normalized entropy vs memorization score.

in this subsection, under a coarser-grained binning of memorization score, EM Linearity is preserved. We conducted a range of coarse-grained memorization scoring experiments where we varied the number of discrete bins from fine-grained (10 bins) to coarse-grained (3 bins).

$n$**-width-binning memorization scoring.** We re-binned (exact) memorization scores into $n_d$ equal-width bins and computed entropy within each bin. For each configuration, we examined both entropy and normalized entropy (relative to the maximum possible entropy for each token's vocabulary size).

**Experimental setup.** We select $n_d \in \{3, 5, 10\}$. The models and datasets follow the choices in the main body, i.e., OLMo-2-1124-7B and its training dataset.

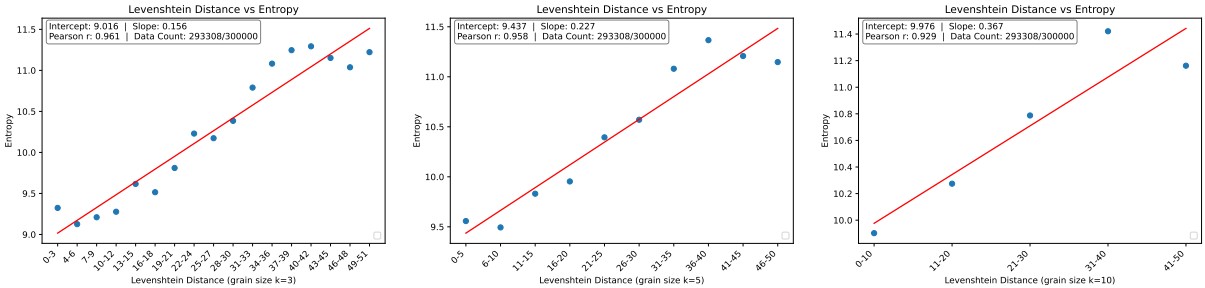

Figure 16: Entropy distributions across memorization bins for different granularities ($n_d = 3, 5, 10$). The core relationship between memorization and entropy is preserved across all binning schemes.

**Results** Figures 16 and 17 show the entropy distributions across different binning granularities. We observe that EM Linearity and Normalized EM Linearity remain preserved across $n_d \in \{3, 5, 10\}$ ($r > 0.9$).

**Conclusion** entropy-memorization relationship is not sensitive to the specific discretization choice.

## C.5  Case study: when does the correlation weaken?

We study when EM Linearity weakens using a case study from Appendix D.1. In Appendix D.1, we segment the sampled data set into 16 clusters with different semantics, and then we test the EM law under these

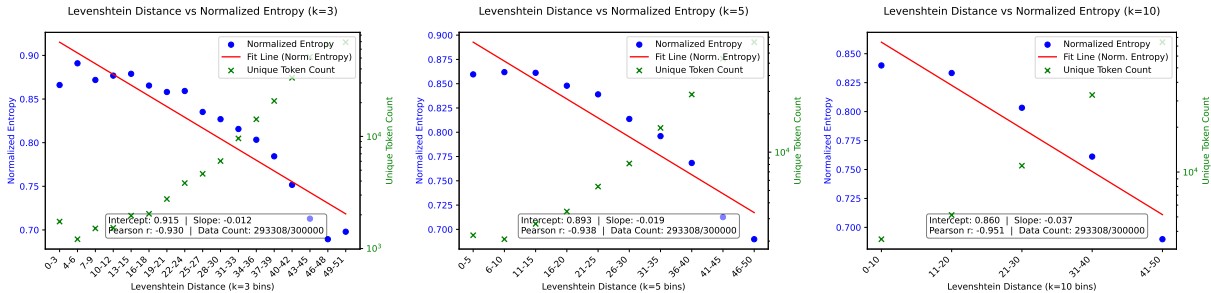

Figure 17: Normalized entropy distributions across memorization bins. Consistent patterns emerge regardless of binning granularity, demonstrating robustness of our findings.

semantically different data. While the relationship remains robust across most domains, we observed that **Cluster 13** ("Genetic and Biomedical Research") exhibited a noticeably weaker correlation ($r = 0.75$). The results are shown in Figure 20n. We suspect that it is induced by a significant domain shift; the corpus in this field is characterized by a unique style of Latin-derived compound words, which deviates from standard natural language. Therefore, we acknowledge that while the EM Relationship is broadly applicable, its predictive power may be more constrained in such out-of-distribution domains.

## C.6 EM Linearity on Instruct Models

In this subsection, we use the level-set-based entropy estimator on the instruct-variant of Olmo 2, OLMo-2-1124-7B-Instruct. The model is trained with supervised fine-tuning (SFT), direct preference optimization (DPO), and reinforcement learning with human feedback (RLHF). For instruct models, we expect memorization to be less pronounced, as explored by Nasr et al. (2025). For example, an LLM that undergoes safety alignment via RLHF (Ouyang et al., 2022) may respond with a generic refusal (e.g., "Sorry, I cannot generate harmful content as a responsible AI.") when prompted with queries that conflict with its safety policies.

Note that considering the purpose and distinct dataset structure in post-training, our "prefix and continuation"-style method is not generally compatible on post-training dataset. Therefore, we tested EM Linearity on the pre-training dataset, in particular, the "DCLM1" subset (around 28,000 samples).

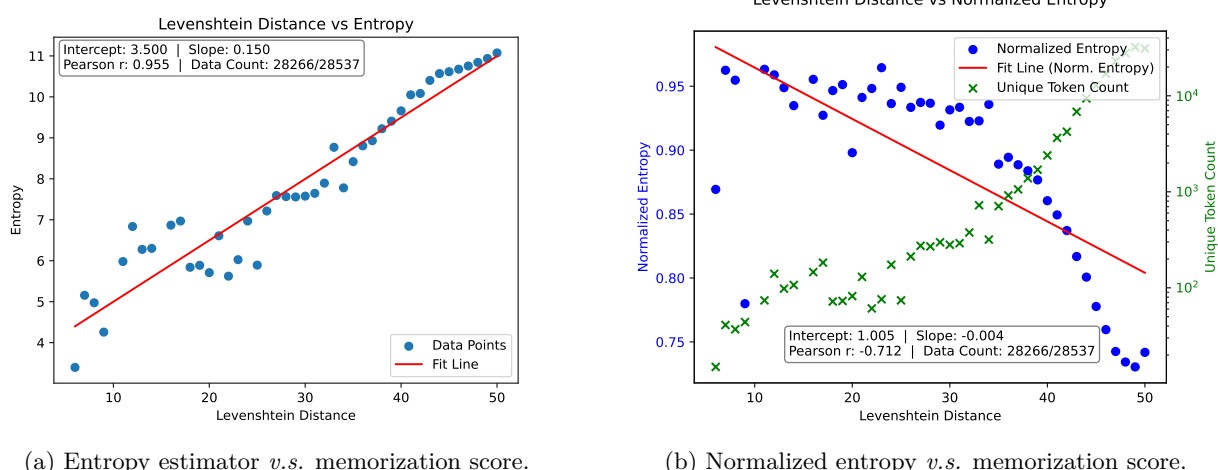

(a) Entropy estimator *v.s.* memorization score.

(b) Normalized entropy *v.s.* memorization score.

Figure 18: Entropy *v.s.* memorization score on instruct models.

The results are presented in Figure 18. In general, measured by edit distance, memorization is markedly reduced in the instruct model. No perfect memorization is detected. The minimum edit distance is 6, consisting of only 15 tokens. Notably, the instruct model also exhibits a robust linear correlation between

entropy and memorization. Furthermore, its regression metrics are remarkably consistent with those of the base model, yielding a comparable intercept (3.500 *v.s.* 3.724), slope (0.150 *v.s.* 0.142), and correlation (0.955 *v.s.* 0.945). In contrast, normalized entropy-memorization correlation is weakened substantially; this is potentially attributable to the reduced token space available in the score $< 10$ regime for low memorization score regimes.

### C.7 Interpret intercept and slope in EM Linearity

Here are some observations on intercept and slope that may shed light on interpretation.

First, for a fixed LLM and its dataset, the intercept and slope are dependent. When the intercept increases, the slope decreases; when the memorization score $e = 50$, $M(s_{50}) \approx 11$. It is evidenced by several experiments along the paper, including experiments with varying generation token lengths (Figure 4), sampling strategies (Figure 7 - 12). Therefore, examining either slope or intercept suffices for our interpretation.

Second, the entropy of the perfect memorization token groups (distance=0) decides the intercept. Evidence is shown in Figure 6 Section 6: the lowest memorization score point exhibits entropy lower than 2, and then the slope in the test data is significantly lower than the slope in the standard train data.

# D   Entropy-Memorization Linearity Under Disparate Semantic Data

We employed a semantic-agonistic strategy to sample the dataset in the main body. This section then explores Entropy–Memorization Linearity under different semantic data. We chunk the sampled dataset into $k = 16$ semantic clusters, develop a strategy to find the semantics of each cluster, and then examine EM Linearity under these disparate semantic data. This experiment was based on the OLMo-1B model using 240,000 sample pieces.

**Semantic Clustering Pipeline**   The steps are as follows:

1. Extracting semantics of token sequences using sentence embeddings. In this step, sentence embeddings project a token sequence to a high-dimensional vector space, where semantically similar sequences are mapped to nearby points. Such embedding techniques are implemented by a twisted version of pre-trained LLM.

2. Clustering. With semantic embeddings, we apply K-Means (Lloyd, 1982) in the latent space and partition the data into $k = 16$ semantic clusters.

3. Identifying semantics of the cluster. Since the clustering methods are performed in a latent space which is not interpretable, we develop a highly-automated pipeline to identify the semantics of each cluster. The core algorithm is differential clustering (Zhang et al., 2025).

4. Run Algorithm 2 on 16 partitions of the dataset. For each cluster, a linear regression is applied. We report the Pearson correlation coefficient, slope, and intercept and visualize the fitted lines.

To implement step 1, we select a popular model *all-mpnet-base-v2* (huggingface, 2025) from the Sentence Transformers library in Huggingface as the encoder. In Appendix D.1, we present how we implement step 3, and provide detailed clustering results with labeled semantics.

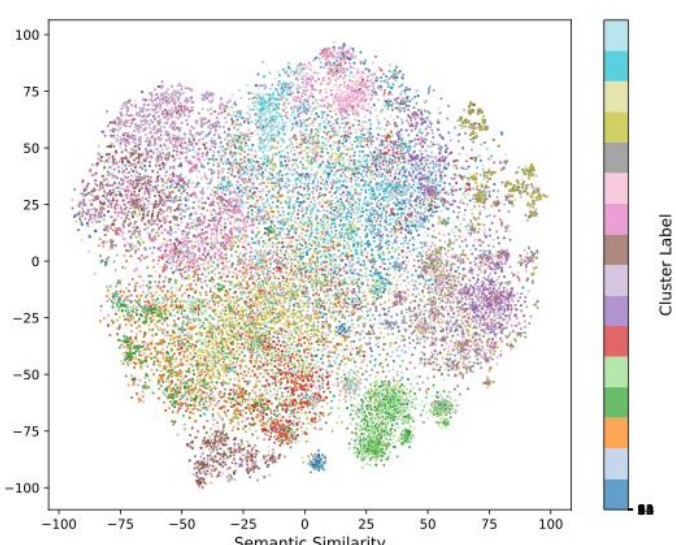

| Cluster | r | Slope | Intercept |
|---|---|---|---|
| 01 – Scientific Research and Technology | 0.82 | 0.04 | 7.86 |
| 02 – Address and Organizational Identifiers | 0.86 | 0.05 | 4.95 |
| 03 – Religious and Biblical Text | 0.91 | 0.05 | 6.56 |
| 04 – Community Engagement and Activities | 0.88 | 0.03 | 8.73 |
| 05 – Social Development and Policy | 0.88 | 0.06 | 7.27 |
| 06 – Software Development Configuration | 0.91 | 0.06 | 6.30 |
| 07 – Spiritual and Religious Beliefs | 0.91 | 0.05 | 6.87 |
| 08 – Health Risks and Medical Studies | 0.89 | 0.08 | 5.77 |
| 09 – Economic Indicators and Growth | 0.86 | 0.08 | 6.23 |
| 10 – Evolutionary Biology and History | 0.86 | 0.03 | 8.45 |
| 11 – Fantasy Narrative (Harry Potter) | 0.83 | 0.05 | 7.25 |
| 12 – Personal Experiences and Emotions | 0.89 | 0.05 | 7.22 |
| 13 – Linguistic and Semantic Analysis | 0.82 | 0.04 | 8.42 |
| 14 – Genetic and Biomedical Research | 0.75 | 0.05 | 7.39 |
| 15 – Programming Command Scripts | 0.87 | 0.02 | 8.93 |
| 16 – Political Events and Commentary | 0.89 | 0.05 | 7.53 |

Figure 19: Clustering sentence embeddings using OLMo-1B pre-training dataset. We apply T-SNE (van der Maaten & Hinton, 2008) for dimension reduction.

**Experimental Results**   Figure 19 presents overall results. It is verified that Entropy–Memorization Linearity is observed among all clusters of data. Moreover, another interesting finding is that, in general,

different clusters exhibit distinctive intercept and slope values. For example, cluster 1 (Address and Organizational Identifiers) exhibits low intercept, while cluster 3 (Community Engagement and Activities) and 14 (Programming) exhibit high intercepts.

> We confirm that Entropy-Memorization Linearity is robust under disparate semantic data clusters. Moreover, intercepts and slopes are different for different semantic data.

### D.1 Technical Details on Interpreting Semantics of Each Cluster

To identify the semantics of each cluster, we build a pipeline that significantly reduces human annotation efforts. The pipeline is as follows:

1. Detect distinctive samples within each cluster. Zhang et al. (2025) formulates this task as a *differential clustering* problem and proposes a FINC method. To quantitatively measure semantic distinctions among the 16 clusters obtained via K-means, we conducted 16 FINC comparisons. For each cluster $C_i$, we set $C_i$ as the novel dataset and the union of the remaining 15 clusters as the reference set. The input to FINC is the sentence embeddings of all instances in the set, and FINC suggests the distinctive samples in the cluster.

2. Keywords summarization. In this stage, we use tri-grams as effective descriptors for naming and interpreting cluster identities. Specifically, we use i) *spaCy* (PyPI, 2025) to perform named entity recognition and dependency parsing to ensure that extracted units are linguistically complete phrases (e.g., *"protective spell harry", "lend broom fly"*), and ii) *YAKE* (Campos et al., 2020) to rank terms using heuristics such as frequency, context, and positional distribution.

3. Human annotation. Based on the summarized keywords, human annotators further summarize the semantics of the cluster.

### D.2 Entropy–Memorization Plot for Each Cluster

Figure 20 presents the detailed plot for each cluster.

### D.3 Interpreting Semantics of Each Cluster

Table 5 presents the top-5 keywords and human-annotated semantic labels for all 16 clusters.

Table 5: Top-5 keywords and human-annotated semantic labels for each of the 16 clusters.

*Cluster 0: Scientific Research and Technology*

| Score | Keyword Phrase |
|---|---|
| 1.28e-04 | translate depth direction |
| 1.28e-04 | design fabrication characterization |
| 1.28e-04 | pct design fabrication |
| 1.28e-04 | manipulation pct design |
| 1.23e-04 | sensor control manipulation |

*Cluster 1: Address and Organizational Identifiers*

| Score | Keyword Phrase |
|---|---|
| 1.74e-05 | street number city |
| 1.54e-05 | party committee number |
| 1.54e-05 | city number |
| 1.52e-05 | type number form |
| 1.46e-05 | conduit state number |

*Cluster 2: Religious and Biblical Texts*

| Score | Keyword Phrase |
|---|---|
| 7.94e-05 | people sword thy |
| 7.94e-05 | thy people sword |
| 7.91e-05 | jacob son reuban |
| 7.84e-05 | son brother house |
| 7.80e-05 | son lord hath |

*Cluster 3: Community Engagement and Activities*

| Score | Keyword Phrase |
|---|---|
| 1.05e-04 | irrigation evaporate leave |
| 1.05e-04 | bullet time jump |
| 1.04e-04 | community meetup world |
| 1.03e-04 | community kid spout |
| 1.02e-04 | year electronic music |

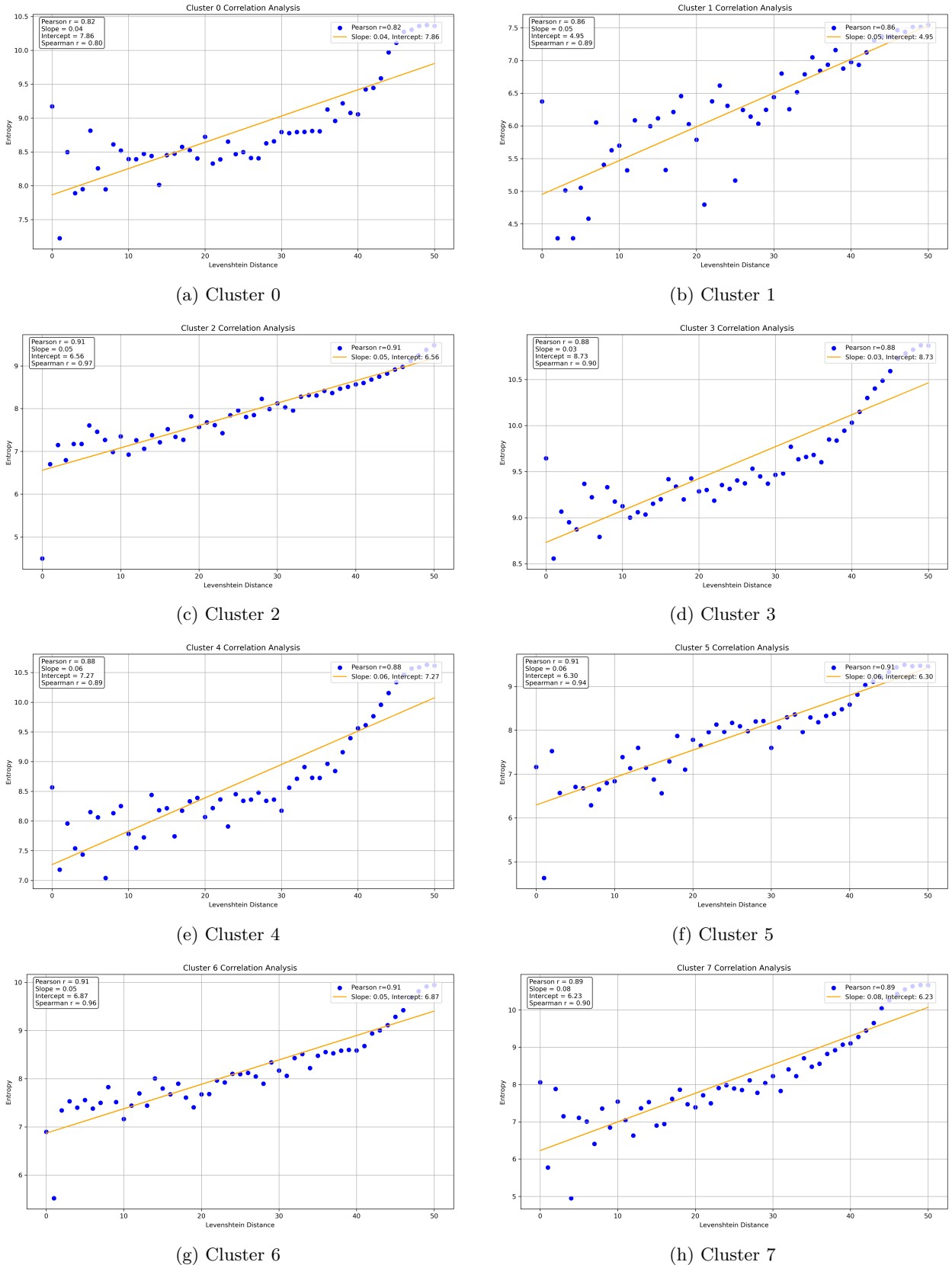

Figure 20: Clusters 0–7.

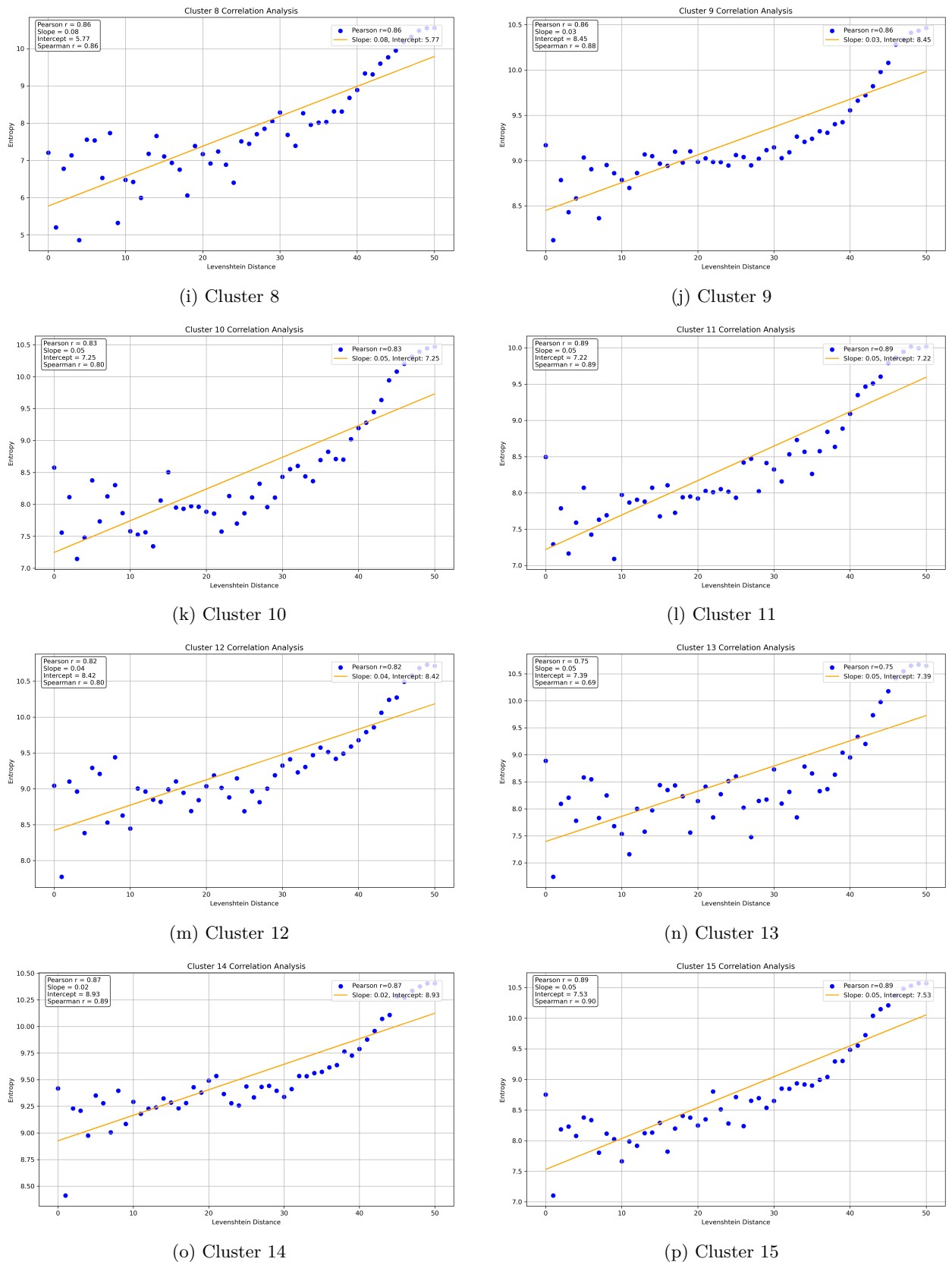

Figure 20: Clusters 8–15.

*Cluster 4: Social Development and Policy*

| Score | Keyword Phrase |
|---|---|
| 1.34e-04 | dramatically drive price |
| 1.33e-04 | develop skill team |
| 1.33e-04 | prefer policy influence |
| 1.33e-04 | outwith prefer policy |
| 1.33e-04 | sector real passion |

*Cluster 5: Software Development Configuration*

| Score | Keyword Phrase |
|---|---|
| 2.39e-05 | true plugin proposal |
| 2.29e-05 | node optional true |
| 2.27e-05 | header content type |
| 2.26e-05 | true ellipsis true |
| 2.26e-05 | dev true child |

*Cluster 6: Spiritual and Religious Beliefs*

| Score | Keyword Phrase |
|---|---|
| 9.67e-05 | thy mind thy |
| 9.61e-05 | death eternal life |
| 9.54e-05 | create sustain universe |
| 9.52e-05 | world drive ulterior |
| 9.52e-05 | behavior world drive |

*Cluster 7: Health Risks and Medical Studies*

| Score | Keyword Phrase |
|---|---|
| 1.06e-04 | health increase risk |
| 1.06e-04 | evaluate risk factor |
| 1.05e-04 | pregnancy increase risk |
| 1.05e-04 | disease high risk |
| 1.05e-04 | high disease risk |

*Cluster 8: Economic Indicators and Growth*

| Score | Keyword Phrase |
|---|---|
| 2.54e-05 | billion country oda |
| 2.51e-05 | permanent surface runway |
| 2.51e-05 | kwh capita growth |
| 2.50e-05 | imf intelsat interpol |
| 2.48e-05 | price growth rate |

*Cluster 9: Evolutionary Biology and History*

| Score | Keyword Phrase |
|---|---|
| 4.49e-04 | primatology million year |
| 4.48e-04 | mutate gene ancient |
| 4.48e-04 | start million million |
| 4.45e-04 | paleolithic million year |
| 4.44e-04 | account million year |

*Cluster 10: Fantasy Narrative (Harry Potter)*

| Score | Keyword Phrase |
|---|---|
| 4.98e-05 | dumbledore harry meet |
| 4.95e-05 | forward harry ron |
| 4.94e-05 | muggle bystander incredulously |
| 4.72e-05 | sirius black peril |
| 4.71e-05 | harbor end time |

*Cluster 11: Personal Experiences and Emotions*

| Score | Keyword Phrase |
|---|---|
| 1.07e-04 | hunt season roll |
| 1.06e-04 | love good love |
| 1.06e-04 | thing work happen |
| 1.05e-04 | decision normal result |
| 1.04e-04 | understand thing happen |

*Cluster 12: Linguistic and Semantic Analysis*

| Score | Keyword Phrase |
|---|---|
| 1.22e-04 | interpret apply male |
| 1.20e-04 | ingen det finne |
| 1.20e-04 | finne ingen det |
| 1.19e-04 | title aktivt medlem |
| 1.17e-04 | confusion attain agenda |

*Cluster 13: Genetic and Biomedical Research*

| Score | Keyword Phrase |
|---|---|
| 8.51e-05 | mesenchymal stem cell |
| 8.40e-05 | screen gene foster |
| 8.27e-05 | interaction show high |
| 8.26e-05 | expression level gene |
| 8.25e-05 | search perform blast |

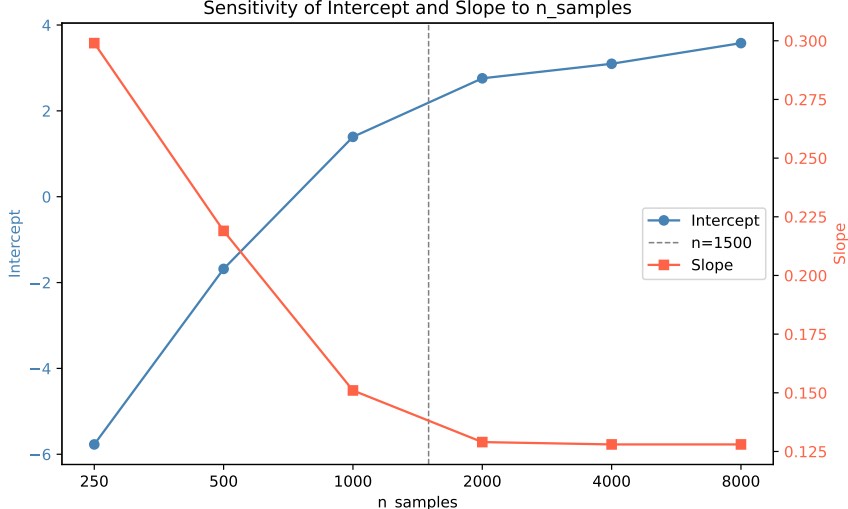

Figure 21: Intercept and slope in EM Linearity change with sample size.

*Cluster 14: Programming Command Scripts*

| Score | Keyword Phrase |
|---|---|
| 4.64e-06 | text text ohm |
| 4.57e-06 | delaytimer command |
| 4.03e-06 | dark text text |
| 3.94e-06 | command runcmd |
| 3.66e-06 | cost command type |

*Cluster 15: Political Events and Commentary*

| Score | Keyword Phrase |
|---|---|
| 1.41e-04 | white house official |
| 1.41e-04 | idea african americans |
| 1.38e-04 | african americans woman |
| 1.38e-04 | end war year |
| 1.37e-04 | snc lavalin scandal |

# E   Extended Results on Dataset Inference

## E.1   Sensitivity test on number of samples required

In practice, we empirically set the minimum number of samples to conduct dataset inference to 1,500. It is motivated by a sensitivity test with the following setup. We sample the OLMo-2-1124-7B dataset (with size 300k) to various sizes $n \in \{250, 500, 1000, 2000, 4000, 8000\}$ to evaluate the stability of the linear relationship between entropy and memorization score. As illustrated in Figure 21, both the intercept and the slope exhibit significant fluctuations at smaller sample sizes ($n < 1,500$). Specifically, when $n$ decreases, the intercept decreases from a positive plateau to $-6$, while the slope undergoes a sharp increase.

Consequently, we select $n = 1,500$ as a principled threshold that balances computational efficiency with statistical reliability, ensuring that our "EM Linearity" parameters are representative of the full data distribution.

## E.2   Comparison with the Baseline Method

**Baseline**   We select the dataset inference performed by Maini et al. (2024) as our baseline method, where the authors propose to learn the weight of membership inference attacks to aggregate these attacks to a final decision. To ensure a fair comparison between Maini et al. (2024) and our method, we use the same amount of data to predict membership. We select 16 metrics in total, suggested by Maini et al. (2024). Key metrics include "perplexity", "k-min probs", "k-max probs", and "zlib ratio". Following the paper, we adopt p-value 0.05 as the decision threshold. To obtain the weights to aggregate MIA results, we sampled 1000 train and validation sentences from the pile. The setting follows the suggestions by the authors.

**Metrics**   In addition to predictive performance, we report the execution time for each algorithm. To ensure a fair comparison, both were evaluated on the same server using a single A100 80GB PCIe GPU.

**Results**   We compare the performance of dataset inference results on Pythia-6.9b-deduped model, and dataset MIMIR, a re-compilation from the Pile.

| Dataset | Ours | Maini (2024)* |
|---|---|---|
| cc_m | 107 | 258 |
| cc_nm | 107 | 257 |
| full_m | 769 | 1532 |
| full_nm | 746 | 1530 |
| tarxiv_m | 121 | 322 |
| tarxiv_nm | 121 | 326 |

Table 6: Runtime comparison across datasets. Beyond a dataset inference run, Maini et al. (2024) requires training regression weights for 810 seconds.

Table 6 compares the time metric. Our method consistently achieves at least 2x speedup compared with Maini et al. (2024). Beyond that, in Maini et al. (2024), training a regression model to obtain weights for aggregation takes 810 seconds. This demonstrates the training-free advantage of our method.

Table 7: Dataset inference prediction comparison.

| LLM | Dataset | Our Pred. | Maini et al. (2024) Pred. | Ground-truth |
|---|---|---|---|---|
| Pythia | MIMIR_cc | 0 | 0 | 0 |
| Pythia | MIMIR_cc | 1 | 0 | 1 |
| Pythia | MIMIR_full | 1 | 0 | 0 |
| Pythia | MIMIR_full | 1 | 0 | 1 |
| Pythia | MIMIR_tarxiv | 1 | 0 | 1 |
| Pythia | MIMIR_tarxiv | 0 | 0 | 0 |

As shown in Table 7, our method achieves 5/6 correct predictions, whereas Maini et al. (2024) achieves 3/6. This shows an improvement in accuracy over the prior approach.

