# OpenReview forum: "Data Compressibility Quantifies LLM Memorization"
_TMLR — Accepted by TMLR_

### Review · Reviewer_XR4f · 2026-01-19

**Summary Of Contributions:**

This paper proposes a quantitative characterization of LLM memorization using data compressibility, and identifies why prior compressibility-based attempts were noisy and weak when computed at the instance level. The main novelty is to introduce a departure from instance-level metrics to set-level (“level-set”) metrics, where samples are grouped by memorization score and compressibility is measured over the aggregated token multiset. With this shift, the paper reports a robust linear relationship between a set-level entropy estimator and memorization (using edit distance), termed the Entropy-Memorization (EM) Law, and report high Pearson correlations across multiple open-source LLM families and experimental settings.

Strengths: (i) the set-level pivot is simple and conceptually clean, (ii) empirical correlations are strong and reproduced across models and settings, (iii) the work provides a quantitative view of memorization, and (iv) it surfaces a practical privacy-auditing style application

Weaknesses / open issues: (i) the “law” is primarily empirical and would benefit from sharper theoretical framing and boundary conditions, (ii) the choice of grouping by exact memorization score is not fully justified relative to other plausible stratifications, and (iii) tokenizer/vocabulary dependence and support-size effects deserve a more explicit discussion of invariances versus parameterization.

**Additional Comments:**

The central insight is strong and refreshingly simple. My main request is conceptual: justify why this notion of “set” is the right unit for defining entropy in this context, and map out a small set of alternative choices so readers understand what is essential versus contingent.

**Audience:**

Yes

**Audience Explanation:**

This will interest people working on (i) memorization/privacy, (ii) dataset inference and membership inference at scale, (iii) empirical “laws” for LLM behavior, and (iv) data-centric understanding of memorization beyond duplication effects. The methodological move is minimal and the result is surprisingly strong, which makes it easy for others to test and potentially incorporate into auditing pipelines.

**Broader Impact Concerns:**

No major concerns on broader impact.

**Claims And Evidence:**

Yes

**Claims Explanation:**

The empirical evidence for a strong linear correlation between set-level entropy and memorization score is convincing. It is demonstrated across several base model families, and the reported correlations are high. The comparison against instance-level compressibility measures (including zlib) supports the core thesis, and it significantly improves the signal-to-noise regime of the statistics.

The caveat is that the paper’s strongest claim is empirical robustness, not universality. I would encourage the authors to phrase “law” as “empirical scaling relationship” unless they can provide a principled argument for when it should or should not hold (e.g., different tokenizations, post-training, heavy domain shifts, extreme prompt regimes).

**Requested Changes:**

- Justify the level-set definition (grouping by memorization score) more explicitly. Right now, the grouping rule is introduced as a practical fix to token-support sparsity, but the paper should articulate the principle: you are conditioning on a target variable (memorization) to estimate a distributional statistic (entropy) in a regime where per-instance support is too small. Also discuss whether exact-score bins are necessary, or whether coarse binning yields the same law.
- Provide intuition/interpretation for the choice of memorization score. Memorization score is defined as edit distance between continuation and ground truth, so “higher memorization score” means less verbatim reproduction. While this is a well accepted measure and the authors cite the paper, but justify why this measure is a good measure for characterization memorization and provide intuition/interpretation.
- Boundary conditions / negative results. Add one controlled setting where the relationship weakens (or a discussion of expected failure modes), to show the authors understand scope.

---

### Review · Reviewer_iL6h · 2026-01-25

**Summary Of Contributions:**

### Summary of Contributions

The authors study the problem of measuring and characterizing memorization in Large Language Models (LLMs) by analyzing the compressibility of training data. They seek to address the limitations of prior research, which failed to find a reliable link between compressibility and memorization since the metrics were evaluated at the instance level, resulting in noisy and unstable estimates.

To overcome this, the authors introduce the **"Entropy-Memorization (EM) Law,"** a set-level approach that groups data by memorization score. This method reveals a robust linear correlation ($r>0.9$) between the entropy of a data set and the model's ability to memorize it.

Additionally, the paper explores the underlying mechanics of this law, showing that perfectly memorized data occupies an exponentially smaller token space. Finally, they leverage these findings to propose a simple, compute-efficient **"Dataset Inference"** method that distinguishes between training and test data for privacy auditing.

### Highlights

* **Robust Methodology:** By shifting from instance-level to set-level analysis, the authors successfully mitigate the statistical noise caused by small token spaces, solving a key issue that hindered previous attempts to link compressibility with memorization.
* **Quantitative and Model-Agnostic:** The proposed entropy estimator provides a quantitative description of memorization rather than a binary one and is model-agnostic, meaning it is compute-efficient and does not require expensive retraining or gradient computation.
* **Extensive Validation:** The EM Law is empirically validated across a wide range of open-source models (including OLMo, OpenLlama, and Pythia), various inference sampling strategies, and disparate semantic clusters.
* **Practical Application:** The research translates the theoretical finding into a practical tool for **Dataset Inference (DI)**, enabling auditors to detect test-set contamination or unauthorized data usage by analyzing the intercept of the entropy-memorization regression line.


### Some Potential Limitations

* **Lack of Granularity:** Due to the reliance on set-level aggregation to ensure statistical stability, the proposed method cannot predict memorization scores at the instance level.
* **Theoretical Gaps:** While the empirical correlations are strong, the authors acknowledge a lack of theoretical justification for certain phenomena, such as the exponential growth of the token space and the specific intercept values observed.
* **Limited Scope:** The study is restricted to pre-trained "base" models and explicitly excludes post-trained variants (like those fine-tuned with RLHF), noting that such training stages induce distribution shifts that complicate the analysis.

**Audience:**

Yes

**Audience Explanation:**

Yes, the findings of this paper are relevant to the TMLR audience as they address open questions in Large Language Model (LLM) research regarding training data dynamics, with implications for both theoretical understanding and practical application.

The specific reasons include:

* **Validation of the Link Between Compressibility and Memorization:** The study provides empirical evidence supporting the theoretical assumption that data compressibility influences memorization. By demonstrating that instance-level metrics are noisy and that a set-level entropy estimator yields a strong linear correlation ($r>0.9$) with memorization scores, the authors offer a methodological solution to quantifying data-centric factors in LLM memorization.
* **Advancement in Memorization Metrics:** Current literature often categorizes memorization as a binary outcome. This work proposes a continuous, quantitative metric based on entropy estimation, allowing for a more granular assessment of how training data distribution affects model generation and privacy risks.
* **Applicability to Privacy and Compliance Auditing:** The proposed "Dataset Inference" method addresses the need for tools to detect test-set contamination and unauthorized use of copyrighted material. The ability to distinguish training data from test data based on regression intercepts provides a practical mechanism for external auditing without requiring access to model weights.
* **Computational Efficiency:** The proposed method is model-agnostic and operates without gradient computation or retraining. This offers a scalable alternative to model-aware approaches, such as influence functions, making it feasible for practitioners to analyze large-scale models and datasets.

**Broader Impact Concerns:**

The submission includes a "Limitations and Broader Impacts" section  that adequately addresses the ethical implications of the work. The authors discuss the positive impacts of their proposed "Dataset Inference" method, framing it as a tool for "privacy auditing" , "pre-screening of training datasets" and potentially further identifying "potential copyright-infringement issues".

**Claims And Evidence:**

Yes

**Claims Explanation:**

Yes, the claims in the submission are well-supported by accurate and convincing empirical evidence, though the theoretical underpinnings remain primarily observational.

### 1. Evidence for the Limitations of Prior Metrics
The authors successfully establish the necessity of their approach by reproducing the failure of existing instance-level metrics. They present scatter plots of zlib compression rates and entropy estimates computed at the instance level, which demonstrate highly noisy distributions and weak Pearson correlations ($r \approx 0.6$). This empirically demonstrates that instance-level metrics are insufficient for accurately quantifying memorization due to the limited sample space of individual sequences.

### 2. Validation of the Entropy-Memorization (EM) Law
* **Strong Statistical Correlations:** By shifting to set-level analysis, the authors achieve Pearson correlation coefficients ($r$) consistently above 0.9 across multiple model families, including OLMo, OpenLlama, and Pythia.
* **Quantitative Visualization:** The regression analyses exhibit clear linear fits, confirming that as the set-level entropy increases, the memorization score (defined by Levenshtein distance) increases linearly.
* **Experimental Robustness:** The authors demonstrate that this linear relationship is not an artifact of specific hyperparameters. The correlation holds ($r > 0.92$) under varying generation lengths (10–50 tokens) and different inference sampling strategies, including Temperature, Top-p, and Top-k.

### 3. Verification of underlying Mechanisms
The authors support their hypothesis regarding the mechanism of the EM Law by analyzing the underlying token space. They provide data showing that the number of unique tokens grows exponentially as the memorization score increases. Specifically, perfectly memorized data (score = 0) is shown to occupy a significantly constrained vocabulary of approximately $2^{10}$ tokens, compared to the much larger vocabulary utilized in non-memorized text. This provides an empirical justification for the link between low-entropy text and high memorization rates.

### 4. Validation of Practical Application (Dataset Inference)
The claim that the EM Law enables effective privacy auditing is supported by successful classification experiments.
* **Distinct Regression Patterns:** The authors demonstrate that test data manifests distinct regression characteristics—specifically different intercepts and steeper slopes compared to training data.
* **Classification Accuracy:** The proposed Dataset Inference method is validated on the "LiveBench" and "MIMIR" datasets. The method successfully distinguishes between member and non-member data by applying a threshold to the regression intercept.

### Limitations of the Evidence
While the empirical evidence is strong, the theoretical derivation is limited. The authors acknowledge they cannot fully explain the exponential growth of the token space or the specific values of the slopes and intercepts derived, suggesting only potential links to Zipf’s law or long-tail distributions. Furthermore, the scope of the evidence is restricted to pre-trained "base" models, as the study explicitly excludes post-trained models (e.g., RLHF) to avoid distribution shifts.

**Requested Changes:**

The current draft of the submission can be substantially strengthened by considering additional discussion on some sections. Some recommendations:

1.  **Robustness Analysis of Dataset Inference Thresholds**
    The current Dataset Inference (DI) method relies on empirically setting a threshold $\tau_k$ (e.g., 0 for Pythia and 3 for OLMo-2). The practical use case suggests determining this threshold using a "known subset of the training data". To validate the robustness of this approach, please consider adding:
    * **Sensitivity Analysis:** Determine the minimum sample size required in the "known subset" to reliably estimate $\tau_k$.
    * **Generalization Checks:** Evidence that a $\tau_k$ derived from one non-member dataset generalizes to others. Currently, the evaluation uses LiveBench and MIMIR. If the threshold is highly sensitive to the specific distribution of the non-member dataset, the claim of a "simple and scalable approach" is significantly weakened.

### Strengthening Changes

1.  **Clarification of "Memorization Score" Terminology**

    The paper defines the "memorization score" as the Levenshtein distance. Consequently, a *higher* score indicates *lower* similarity (poorer memorization). This nomenclature is counter-intuitive. Consider renaming this metric to **"Memorization Error"**, **"Generation Divergence"**, or simply **"Edit Distance"** throughout the text to improve readability and prevent misinterpretation of the results.

2.  **Validation on Instruction-Tuned Models**
    The study explicitly excludes post-trained models (RLHF/SFT) due to potential distribution shifts. While this scope limitation is noted, testing the EM Law on at least one instruction-tuned variant (e.g., `OLMo-Instruct` or `Pythia-Chat`) would strengthen the work.
    * If the law breaks, it empirically confirms the hypothesis regarding distribution shifts.
    * If the law holds, it expands the applicability of the findings to the model types most frequently deployed in practice.

3.  **Comparisons with Membership Inference Baselines**
    To contextualize the effectiveness of the EM Law-based auditing, it would be beneficial to compare its performance against standard reference-free membership inference baselines (e.g., Min-K% Prob, perplexity-based thresholding) on the same subsets. This comparison would clarify whether the set-level entropy method offers superior discriminative power or primarily provides a computational advantage.

4.  **Theoretical Discussion on Slope and Intercept**
    The paper notes that the regression intercept and slope vary by semantic cluster, but offers limited theoretical justification beyond a reference to Zipf's law. Expanding the discussion to include a hypothesis or simulation regarding *why* specific semantic categories (e.g., "Address and Organizational Identifiers" vs. "Fantasy Narrative") yield such distinct parameters would add theoretical depth to the empirical observations.

---

### Review · Reviewer_WsgX · 2026-02-12

**Summary Of Contributions:**

The paper investigates the relationship between the entropy/compressibility of the data and the degree to which Large Language Models (LLMs) memorize that data. The authors argue that previous research has been "instance-level" (looking at individual sentences), making it noisy and failing to provide a clear quantitative "characterization" or universal rule for why some data is memorized more than others. The paper proposed the Level-set-based Entropy Estimator to better understand the memorization in LLMs, and discovers the EM Law that the set-level data entropy is linearly correlated with the model's memorization score. Moreover, the paper observes that highly memorized data (low edit distance) consists of significantly fewer unique tokens, suggesting lower lexical diversity. The paper also studied whether the behavior could be transfered to the test dataset, and studied Dataset Inference as a potential application of the proposed metric.

**Strengths**
1. The proposed method is model-agnostic and training-free, only requiring the model's output and the data. This makes it computationally cheaper than methods requiring gradients or influence functions.

2. It is novel to identify the low diversity in instance-level analysis and extended the method to set-level analysis.

3. The authors validated the correlation across several open-source model families (Pythia, OLMo, TinyLlama), suggesting the phenomenon is not restricted to a single architecture.

**Weaknesses**

1. The set $\mathcal{T}_e$ is itself defined with the Levenshtein distance. Based on $\mathcal{T}\_e$, the paper defined the empirical entropy. Therefore, the EM Law's claim that the empirical entropy correlated well with the Levenshtein distance is a circular argument and does not have significant meaning.

2. In the definition of the empirical distribution, the normalizer $N|s|$ seems problematic as the probability does not sum to be 1.

3. The limitation of the set-level analysis llies in the fact that it cannot tell a user if one specific document (like a private medical record) was memorized, which is the primary concern in privacy auditing.

**Audience:**

Yes

**Audience Explanation:**

The topic of LLM memorization is important both to the understanding of the mechanism of LLMs and to resolve the safety and privacy concerns in the deployment of the model.

**Claims And Evidence:**

No

**Claims Explanation:**

The paper claims that "existing LLM memorization research has o!ered limited insight into how training data influences memorization and largely lacks quantitative characterization". However, it seems that the paper "Quantifying Memorization Across Neural Language Models" by Carlini et al. offers such quantitative characterization. Similarly the paper "Quantifying and Analyzing Entity-level Memorization in Large Language Models" by Zhou et al.

The paper claims a "desirable accuracy" on the DI task, and is "is compute-effcient", but provides no comparison of accuracy and runtime with the baseline methods.

**Requested Changes:**

1. Clarify on the definition of set-level entropy and Levenshtein, and explain why the EM Law is significant.

2. Justify the normalizer in the definition of the empirical distribution in Eq. (5).

3. Give a better discussion about the papers on LLM memorization mentioned above.

4. Provide comparison with DI baseline methods.

---

### Decision · Action_Editor_35wr · 2026-04-02

**Recommendation:** Accept with minor revision

**Additional Comments:**

The authors need to provide a camera-ready version of the paper with all of the promised additions/changes.

**Audience:**

Yes

**Audience Explanation:**

This paper deals with interesting and important questions related to memorization in LLMs, an important topic in the ML community.

**Claims And Evidence:**

Yes

**Claims Explanation:**

In this paper the authors identify a robust "Entropy-Memorization (EM) Linearity," which demonstrates a clear linear correlation between data entropy estimators and LLM memorization scores. They claim to validate this linearity  across numerous open-source models and offer a practical method for distinguishing between an LLM's training and test data.

In the first set of reviews, the reviewers had a split consensus on whether the paper's claims are backed by clear and convincing evidence. Two reviewers praised the robust empirical validation across multiple models, noting strong statistical correlations, though they cautioned that the findings were primarily observational and lacked deep theoretical grounding. However, a third reviewer initially found the evidence unconvincing, pointing out potential circular logic in the metric's formulation, problematic normalizers, and missing baseline comparisons. Ultimately, while the empirical results were compelling, the reviewers indicated that the paper required tighter theoretical justification and stronger comparative analyses to fully support its claims.

Following rebuttal, the reviewers were largely satisfied with the response from the authors, and felt that the claims were now well supported. The result was that all three reviewers recommended acceptance.